# Multimodal Fact-Level Attribution for Verifiable Reasoning

**David Wan** [*1] **Han Wang** [*1] **Ziyang Wang** [1] **Elias Stengel-Eskin** [2] **Hyunji Lee** [1] **Mohit Bansal** [1]

## Abstract

Multimodal large language models (MLLMs) are increasingly used for real-world tasks involving multi-step reasoning and long-form generation, where reliability requires grounding model outputs in heterogeneous input sources and verifying individual factual claims. However, existing multimodal grounding benchmarks and evaluation methods focus on simplified, observation-based scenarios or limited modalities and fail to assess attribution in complex multimodal reasoning. We introduce MURGAT (**Mu**ltimodal **R**easoning with **G**rounded **At**tribution), a benchmark for evaluating fact-level multimodal attribution in settings that require reasoning beyond direct observation. Given inputs spanning video, audio, and other modalities, MURGAT requires models to generate answers with explicit reasoning and precise citations, where each citation specifies both modality and temporal segments. To enable reliable assessment, we introduce an automatic evaluation framework that strongly correlates with human judgments. Benchmarking with human and automated scores reveals that even strong MLLMs frequently hallucinate citations despite correct reasoning. Moreover, we observe a key trade-off: increasing reasoning depth or enforcing structured grounding often degrades accuracy, highlighting a significant gap between internal reasoning and verifiable attribution. Code and data are available at https://github.com/meetdavidwan/murgat.

## 1. Introduction

Reliable and trustworthy real-world deployment of multimodal large language models (MLLMs) requires outputs that are verifiable and grounded in a model's input sources. This grounding is particularly important when problems require multi-step reasoning, which amplifies the risk of hallucinations from propagating errors (Ji et al., 2023; Min et al., 2023), and when producing long-form responses which are harder and more time-consuming to verify (Song et al., 2025; Li et al., 2022). While prior work in temporal video grounding (Hendricks et al., 2017; Lei et al., 2021) and multimodal retrieval-augmented generation (Dong et al., 2025; Yu et al., 2025; Chen et al., 2022) has explored grounding multimodal models' outputs to their inputs using citations or timestamps, existing studies often focus on *simplified* settings. Many grounding tasks emphasize *observational or retrieval-based grounding*, where questions can be answered by directly grounding to relevant evidence in the input source (e.g., *"How many flags are in front of the U.S. Capitol dome?"* given an image of the Capitol). In contrast, real-world questions frequently require not only grounding to evidence, but also reasoning over grounded information to synthesize an answer (e.g., the question in Figure 1). Moreover, prior work is typically limited to a *narrow set of modalities*, most commonly visual information. Even in video grounding settings (Hendricks et al., 2017; Wang et al., 2025a; Lei et al., 2019; 2021), existing methods mostly ground to visual inputs or rely on automatically-transcribed text rather than original audio, overlooking modalities such as audio and figures and failing to evaluate joint grounding across heterogeneous multimodal sources.

To evaluate MLLMs in more realistic settings requiring reasoning grounded in heterogeneous multimodal inputs, we introduce Multimodal Reasoning with Grounded Attribution (MURGAT). We measure different models' ability to perform fact-level multimodal attribution in settings that require reasoning beyond direct observation. As shown in Figure 1 (top), given multimodal inputs including video, audio, and graphs, models should generate answers with *explicit reasoning* and precise citations that refer to the *specific modality and temporal segments* supporting each claim. To assess a model's ability to identify and attribute supporting evidence, we decompose response evaluation into three subtasks (Figure 1, bottom). (1) **Verifiable claim identification** identifies sentences that contain directly observable claims requiring grounding, as opposed to sentences that reflect reasoning steps. This allows attribution quality to be

---
*Equal contribution [1]UNC Chapel Hill [2]The University of Texas at Austin. Correspondence to: David Wan <davidwan@cs.unc.edu>.

*Proceedings of the 43rd International Conference on Machine Learning*, Seoul, South Korea. PMLR 306, 2026. Copyright 2026 by the author(s).

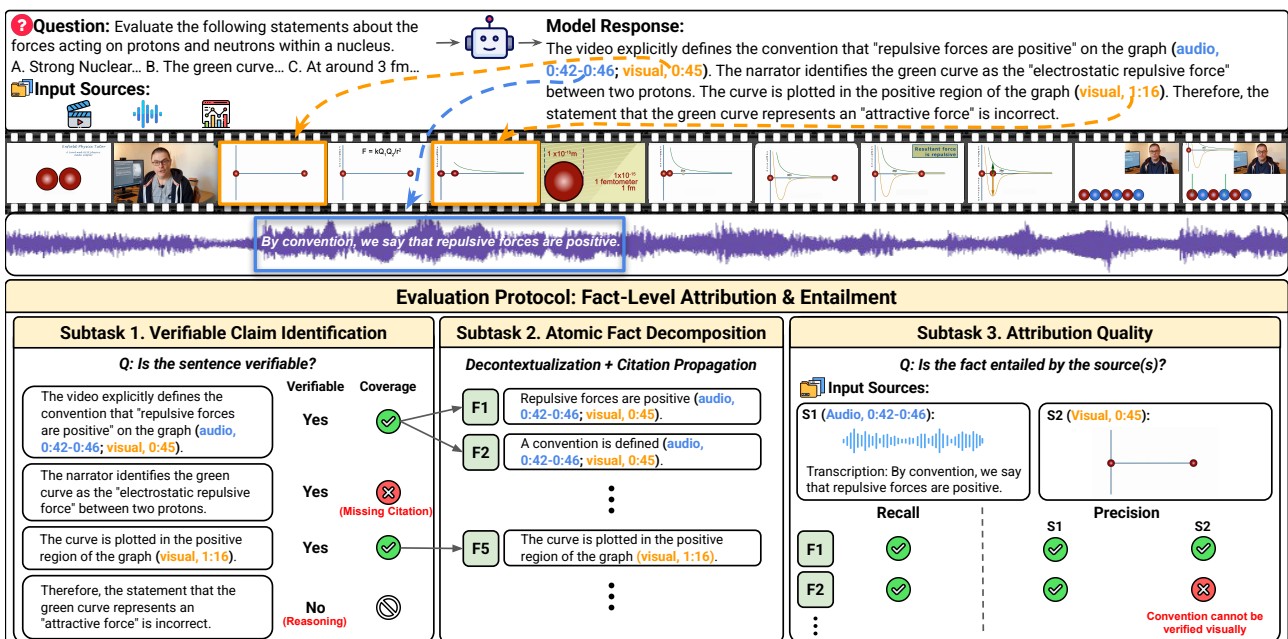

*Figure 1.* Overview of MᴜRGᴀᴛ and the evaluation protocol. The model is given a question and multimodal sources and is asked to generate a response containing explicit reasoning and precise citations, including the specific modality and timestamp. To evaluate the response, we apply a fact-level multimodal attribution protocol. The generated response and its citations are processed through three subtasks: (1) verifiable claim identification, (2) atomic fact decomposition, and (3) attribution quality.

evaluated only over verifiable claims without penalizing ungrounded reasoning or rewarding unnecessary citations. (2) **Atomic fact decomposition** further breaks each verifiable sentence into atomic facts, enabling fine-grained evaluation, as a single sentence often contains multiple claims (Min et al., 2023). (3) **Attribution quality** evaluates whether each atomic fact is entailed by the multimodal evidence cited for it. Following text attribution work (Gao et al., 2023b), we measure recall (whether the union of cited segments fully entails the fact) and precision (whether each cited segment is strictly necessary) while accounting for both temporal alignment and modality.

To establish reliable reference points for these tasks, we first collect human annotations for all three subtasks of the evaluation pipeline on two datasets, WorldSense (Hong et al., 2025) and Video-MMMU (Hu et al., 2025b), which cover a diverse range of multimodal inputs and question types, including those requiring reasoning beyond direct observation. Using these annotations as ground truth, we evaluate a range of MLLM variants (e.g., Gemini-2.5-Flash (Comanici et al., 2025), Gemini-3-Flash and Pro (Google, 2025), and Qwen3-Omni-Instruct and Thinking (Xu et al., 2025)). We observe that even strong MLLMs perform poorly on the MᴜRGᴀᴛ task (Table 1): while they are often able to answer questions correctly, they frequently fail to provide sufficient and accurate attribution to the underlying sources. These findings motivate the construction of an automatic, scalable evaluation pipeline MᴜRGᴀᴛ-Sᴄᴏʀᴇ to efficiently benchmark

methods and improve attribution ability. We experiment with various strong MLLMs to identify the model with the highest correlation to human judgments for each task, and observe a high Pearson correlation $r$ of 0.84 when averaged over all steps, substantially outperforming the next-best LLM-as-judge baseline ($r = 0.59$).

With MᴜRGᴀᴛ-Sᴄᴏʀᴇ, we test state-of-the-art MLLMs, including Gemini models and Qwen3-Omni variants. Our experiments reveal that while these models often achieve high question-answering accuracy, they struggle significantly with multimodal attribution, frequently producing "hallucinated grounding" where incorrect citations are given. We specifically observe that citation generation is task-dependent: it acts as a "reasoning tax" (Zhang et al., 2025; Wan et al., 2025) on simple recognition tasks but scaffolds performance on complex reasoning benchmarks. We further explore programmatic approaches that decouple reasoning from citation generation. While these methods improve attribution quality (avg. +9.6 MᴜRGᴀᴛ-Sᴄᴏʀᴇ), we observe a distinct trade-off: forcing explicit grounding often degrades reasoning performance in complex tasks. Finally, we investigate the effect of scaling thinking effort and observe diverging trends: while larger models (e.g., Gemini-3-Pro) improve in grounding with more compute, smaller models show a drop in MᴜRGᴀᴛ-Sᴄᴏʀᴇ as effort increases, suggesting latent reasoning processes become disconnected from verifiable evidence.

## 2. Related Work

**Attribution and Grounding Benchmark.** In text domains, prior work (Bohnet et al., 2022; Jacovi et al., 2025; Gao et al., 2023b; Yue et al., 2023; Li et al., 2024) has studied attribution and grounding as mechanisms to mitigate hallucinations and improve the trustworthiness of model outputs by introducing metrics and benchmarks for evaluating citation quality. Several lines of work propose decomposing outputs into atomic facts for finer-grained evaluation, as sentences often contain multiple factual claims (Min et al., 2023; Wei et al., 2024; Lee et al., 2024). In the multimodal domain, grounding is commonly framed as referring text to specific visual or temporal evidence. Several works evaluate citation generation from MLLMs over visual content. MCiteBench (Hu et al., 2025a) and MAVIS (Song et al., 2025) target image-based VQA with document-level evidence, leaving the temporal and audio modalities unaddressed. MIRAGE (Martin et al., 2025) converges on a similar atomic-decomposition and VLM-verification pipeline for multimodal RAG and likewise finds that strong models frequently hallucinate citations even when their answers are correct. MURGAT differs along two axes: (1) it requires fine-grained *temporal and per-modality* citations rather than source-level attribution to a whole video or document; (2) it explicitly distinguishes verifiable claims from reasoning steps, enabling evaluation of multi-step reasoning responses rather than treating every sub-claim as observable. Video grounding tasks, which aim to localize a relevant segment given a textual query (Hendricks et al., 2017; Lei et al., 2021; Xiao et al., 2024), are also related. Existing methods (Ren et al., 2023; Huang et al., 2024; Wang et al., 2025a;b) assume that the target evidence is already specified in the prompt. In our setting, the model must self-select evidence, rather than selecting a timestamp provided in the prompt.

**Attribution and Grounding Methods.** In text domains, existing attribution approaches mainly fall into three groups: (1) Direct generation approaches (Weller et al., 2024) use attribution from parametric knowledge by prompting language models to cite supporting sources during generation. (2) Post-retrieval attribution methods (Nakano et al., 2021; Menick et al., 2022; Asai et al., 2024) incorporate an explicit evidence retrieval step and enable citation-aware reasoning. (3) Post-generation attribution methods (Gao et al., 2023a; Chen et al., 2024; Hsu et al., 2024) verify or revise claims after the response is produced. Meanwhile, multimodal grounding has become a focus in tasks like long video QA (Wang et al., 2025e;d), where models must locate visual segments to support answers. Recent efforts (Wang et al., 2025c; Mahmood et al., 2025; Wang et al., 2025f; Li et al., 2026) propose programmatic or agent-based reasoning frameworks that decompose queries into executable steps over video content. Modular reasoning approaches structure multimodal inference through specialized sub-modules

or visual programs to improve temporal grounding and interpretability (Surís et al., 2023; Min et al., 2024). These methods improve multimodal retrieval and reasoning but typically focus on answer accuracy or temporal localization over fine-grained attribution of generated claims.

## 3. Task and Evaluation

In this section, we present MURGAT (Multimodal Reasoning with Grounded Attribution) in Section 3.1 and describe the evaluation protocol for measuring model performance in Section 3.2.

### 3.1. MURGAT

As illustrated in the top panel of Figure 1, MURGAT is a task in which an MLLM is given multimodal inputs $I$ from various modalities (e.g., video, audio stream, or figures) and a question $Q$. The model produces a response $R = \text{MLLM}(Q, I)$ consisting of a sequence of sentences $\{r_i\}$ with explicit reasoning. For each verifiable sentence $r_i$ (i.e., a sentence that is observable from the input source $I$) we require the model to generate an associated citation set $C_i = \{c_i^1, c_i^2, \dots\}$, where each citation $c_i^j$ refers to a specific timestamped segment of a particular input modality (e.g., (audio, 0:42-0:46)). If $|C_i| = \emptyset$, the sentence $r_i$ is not accompanied by any citation. We require that all verifiable claims be supported by citations, and that each claim be strictly entailed by the cited sources.

### 3.2. Evaluation Protocol

As shown in the bottom panel of Figure 1, to evaluate MURGAT, we introduce an evaluation protocol with three subtasks: identifying verifiable sentences in the response, decomposing them into atomic facts, and evaluating citation quality of these facts. Based on this protocol, we define an evaluation metric, MURGAT-SCORE (MURGAT-S), which measures how well a model grounds factual claims to the correct source at the fact-level, without incorrectly penalizing unobservable sentences such as reasoning statements.

**Subtask 1: Verifiable Claim Identification.**

In this subtask, the goal is to identify which sentences in a generated response $R$ are verifiable. This process consists of two stages. First, for each sentence $r_i \in R$, we prompt an LLM-based verifier to determine whether the sentence is *verifiable* (Liu et al., 2023a), i.e., whether its claims can be grounded to the multimodal inputs $I$. This yields a filtered set of verifiable sentences: $R_v = \{r_i \in R \mid \text{Verifier}(r_i, I) = \text{True}\}$. This step distinguishes sentences that are visually or audibly verifiable from those that cannot be grounded (Liu et al., 2023a). We retain sentences describing observable events (visual actions, audio, on-screen text). Conversely, we discard sentences that cannot be directly

grounded, such as reasoning statements. For example, in Step 1 of Figure 1, the final sentence (*"Therefore, the statement that · · · is incorrect"*) is filtered out as it reflects reasoning rather than observable evidence. In contrast, the first sentence (*"The video explicitly defines · · · on the graph"*) is retained, since the definition can be directly observed from the video. Second, among the set of verifiable sentences $R_v$, we further retain only those with non-empty citation sets ($C_i \neq \emptyset$) and obtain a set of verifiable and citation-covered sentences $R_{vc} = \{r_i \in R_v \mid C_i \neq \emptyset\}$. These sentences are subsequently passed to the decomposition and attribution subtasks, since attribution quality can only be evaluated when citations are provided.

**Subtask 2: Atomic Fact Decomposition.** A single sentence often contains multiple facts, thus containing a mixture of true and false information (Min et al., 2023). To enable fine-grained evaluation, we decompose each sentence $r_i \in R_{vc}$ into a set of atomic facts $A_i = \{a_i^1, a_i^2, \ldots, a_i^n\}$, where each atomic fact represents a minimal, independently verifiable claim. To ensure accurate evaluation, we apply decontextualization (Choi et al., 2021; Wei et al., 2024), where pronouns are resolved to specific entities using the preceding context. Additionally, unlike prior work (Choi et al., 2021; Wei et al., 2024), since our task involves citations, we propagate the citation set $C_i$ associated with each original sentence $r_i$ to all atomic facts derived from it, yielding atomic fact-citation pairs $\{(a_i^j, C_i)\}$ for subsequent attribution evaluation.

**Subtask 3: Attribution Quality.** Given the atomic fact-citation pairs $\{(a_i^j, C_i)\}$, we evaluate the entailment of each atomic fact with respect to its cited sources. We adopt a set-based evaluation protocol used in prior work (Liu et al., 2023a; Gao et al., 2023b). For each verifiable atomic fact $a_i^j$ and its citation set $C_i$, we perform a two-sided citation verification. First, we determine whether the combination of all citation segments $c_i^k \in C_i$ fully entails the fact $a_i^j$ (*Recall*). This measures whether the provided citations are sufficient to support the fact. Second, if the atomic fact is supported, we further identify which specific citations $c_i^k \in C_i$ are strictly necessary for entailment (*Precision*). This assesses whether each cited segment contributes relevant evidence or whether spurious or overly broad citations are included. Together, these two criteria characterize the quality of multimodal attribution by capturing both missing citations and incorrect or unnecessary citations.

## 3.3. Evaluation Metrics

We propose evaluating grounding quality along two primary axes: **Coverage** and **Attribution**. We then combine these into a holistic score, MURGAT-SCORE.

### 3.3.1. CITATION COVERAGE

Citation coverage measures the model's ability to correctly provide citations for sentences that require grounding. Specifically, it is defined as the proportion of verifiable sentences that are accompanied by at least one citation:

$$\text{Coverage (\%)} = \frac{|R_{vc}|}{|R_v|} \times 100$$

A high score indicates that the model consistently provides attributions for verifiable content.

### 3.3.2. ATTRIBUTION QUALITY

Attribution quality evaluates whether the cited multimodal evidence correctly supports each atomic fact. For the set of all atomic facts $A = \bigcup_{r_i \in R_{vc}} A_i$, we evaluate the quality of evidence using Precision, Recall, and F1, similar to the definitions in Liu et al. (2023a); Gao et al. (2023b).

**Precision:** Assesses the relevance of citations. For a given response, we calculate precision by pooling all citations found in that response. A citation $c_i^j \in C_i$ is considered "relevant" if it supports the associated atomic fact.

$$\text{Precision} = \frac{\sum_{a_i^j \in A} \sum_{c_i^k \in C_i} \mathbb{I}(c_i^k \text{ is relevant})}{\sum_{a_i^j \in A} |C_i|}$$

**Recall:** Measures the sufficiency of the provided evidence. It is the percentage of atomic facts where the set of citation $C_i$ *fully entails* the fact $a_i^j$.

$$\text{Recall} = \frac{1}{|A|} \sum_{a_i^j \in A} \mathbb{I}(C_i \text{ fully supports } a_i^j)$$

**Attribution F1:** We derive the final attribution score (Attribution) as the harmonic mean of Precision and Recall.

$$\text{Attribution} = 2 \cdot \frac{\text{Precision} \cdot \text{Recall}}{\text{Precision} + \text{Recall}}$$

### 3.3.3. MURGAT-SCORE

To provide a single holistic score, MURGAT-SCORE scales the attribution quality by the coverage. This penalizes models that hallucinate citations for a small subset of facts while leaving the majority ungrounded.

$$\text{MURGAT-SCORE} = \text{Coverage} \times \text{Attribution}$$

## 4. Automatic Evaluation

In this section, we describe how we design and identify an *automated* evaluation pipeline for scalable and efficient benchmarking by prompting different models to simulate

*Table 1.* Model performance based on human annotations, where MURGAT-SCORE is computed using annotator labels.

| Model | WorldSense | | | | Video-MMMU | | | |
| --- | --- | --- | --- | --- | --- | --- | --- | --- |
| | Coverage | Attribution | MURGAT-S | Acc | Coverage | Attribution | MURGAT-S | Acc |
| Qwen3-Omni-Instruct | 55.1 | 35.4 | 27.3 | 56.0 | 35.9 | 14.9 | 5.6 | 67.4 |
| Qwen3-Omni-Thinking | 47.1 | 41.2 | 23.1 | 56.0 | 45.0 | 23.4 | **21.8** | 76.0 |
| Gemini-2.5-Flash | **85.0** | **65.8** | **59.9** | 58.0 | 57.1 | **32.5** | **21.8** | 72.0 |
| Gemini-3-Pro | 76.8 | 61.6 | 49.7 | **60.0** | **59.7** | 24.8 | 16.3 | **86.0** |

the three annotation steps and showing high correlations with human judgments (Section 4.2, Section 4.3, and Section 4.4). Finally, using the best methods from the individual subtasks, we develop the final metric, MURGAT-SCORE and evaluate in an end-to-end manner in Section 4.5. Before introducing an automated metric, we construct a sample of human annotations and evaluate current MLLMs with the human annotations samples (Section 4.1).

## 4.1. Human Annotation

To benchmark models and validate automatic metrics, we first constructed human annotations for all three stages of the evaluation pipeline. To capture diverse model behaviors, we randomly sampled 10 examples from each of two datasets: Video-MMMU (Hu et al., 2025b) and WorldSense (Hong et al., 2025), both of which feature multimodal inputs and complex queries. We elicit human judgments on outputs from four widely used and strong MLLMs, *Gemini-2.5-Flash* (Comanici et al., 2025), *Gemini-3-Pro* (Google, 2025), *Qwen3-Omni-Instruct* (Xu et al., 2025), and *Qwen3-Omni-Thinking* (Xu et al., 2025), on the sampled 20 examples, yielding 80 model-generated responses. Human annotators are provided with input sources and model-generated responses, along with stage-specific instructions. More details of human annotation are in Appendix A.

**Results.** Before introducing an automated metric, we report the scores human annotators gave to each model on the sampled videos from WorldSense and Video-MMMU, reporting our evaluation metrics (Coverage, Attribution F1) as well as the QA accuracy of each model. In Table 1, we find that models are far from ceiling performance in terms of coverage and attribution F1, with inconsistent trends between models and no single model performing consistently on both datasets. Moreover, scores on Video-MMMU – which requires detailed grounding to complex visual sources like plots – are generally lower than those on WorldSense, despite the QA accuracy scores being higher. These results underscore the challenge this task poses to even strong MLLMs, and highlight the need for an automated and more scalable evaluation method. Qualitative analysis further suggests a fundamental trade-off between narrative synthesis and grounding precision; while larger models often hallucinate spatial or temporal details to maintain narrative

*Table 2.* Evaluation results for verifiable claim identification (*Subtask 1*) and attribution quality (*Subtask 3*).

| Model | Format | Verifiable | Attribution Quality | | |
| --- | --- | --- | --- | --- | --- |
| | | BAcc | Prec. | Rec. | F1 |
| Gemini-2.5-Flash | Simple | 78.0 | 72.9 | 72.9 | 72.9 |
| | CoT | 75.8 | 70.0 | 70.6 | 70.3 |
| | JSON | 80.6 | 72.1 | 71.4 | 71.7 |
| Gemini-3-Flash | Simple | 80.8 | 65.1 | 66.5 | 65.8 |
| | CoT | 80.2 | 65.0 | 66.2 | 65.6 |
| | JSON | 81.1 | 63.6 | 63.8 | 63.7 |
| Gemini-3-Pro | Simple | 79.0 | 69.3 | 70.3 | 69.8 |
| | CoT | 81.4 | 71.2 | 72.1 | 71.7 |
| | JSON | **84.2** | **72.8** | **73.5** | **73.1** |

fluency, models like Gemini-2.5-Flash achieve higher faithfulness through minimalist, shot-by-shot descriptions (see Section D.2 for a detailed case study and examples).

## 4.2. Subtask 1: Verifiable Claim Identification

**Dataset.** From our annotated data, we collate a sentence-level dataset of 580 examples, each consisting of a sentence paired with a human label indicating its verifiability.

**Methods.** We evaluate three different models using three distinct prompting styles, following Jacovi et al. (2025): a *Simple* prompt that directly outputs a binary decision; a *Chain-of-Thought (CoT)* prompt that requests reasoning before the answer; and a *JSON* structured prompt, a structured variant of CoT that enforces a schema requiring reasoning prior to the verdict which is identified by Jacovi et al. (2025) as a top-performing method. Prompts are in Appendix E and results on additional models can be found in Appendix B.1.

**Metric.** As this task involves a binary decision, we evaluate performance based on Balanced Accuracy (BAcc), a standard practice for unbalanced labels (Laban et al., 2022).

**Results.** The results are presented in Table 2. We observe that Gemini-3-Pro with the JSON prompt achieves the highest performance (84.2 BAcc), with the same model's CoT version performing the next best (81.4 BAcc).

*Table 3.* Correlation results for atomic fact decomposition (*Subtask 2*) on Gemini models, reporting F1 and Citation Propagation (Cit. Prop.). We compare the full (*Full*) pipeline against ablations without decontextualization (*w/o Decontext.*) and a combined single-pass generation (*Single Pass*).

| Model | Format | Sentence-level | | Response-level | |
|---|---|---|---|---|---|
| | | F1 | Cit. Prop. | F1 | Cit. Prop. |
| Gemini-2.5-Flash | Full | 81.0 | 85.5 | 77.8 | 79.9 |
| | w/o Decontext. | 78.7 | 84.2 | 77.3 | 78.2 |
| | Single Pass | 77.5 | 81.6 | 78.4 | 80.5 |
| Gemini-3-Flash | Full | 81.4 | 85.3 | 79.7 | 81.4 |
| | w/o Decontext. | 79.0 | 84.0 | 78.5 | 81.9 |
| | Single Pass | 77.7 | 82.7 | 77.8 | 80.0 |
| Gemini-3-Pro | Full | **81.8** | **86.4** | **80.1** | **84.7** |
| | w/o Decontext. | 79.8 | 85.2 | 79.0 | 84.0 |
| | Single Pass | 78.8 | 83.9 | 79.7 | 82.7 |

*Table 4.* Correlation of metrics with human judgments. We report Pearson ($r$) coefficients across Coverage, Attribution Precision, Attribution Recall, and MURGAT-SCORE. **Ours** is obtained by our evaluation protocol. Dis. is Disentangled. Best results are **bolded**.

| | Coverage | Attr. Precision | Attr. Recall | MURGAT-SCORE |
|---|---|---|---|---|
| Holistic | 0.38 | 0.39 | 0.43 | 0.35 |
| Dis. | 0.58 | 0.32 | 0.49 | 0.45 |
| Dis. (sent) | 0.76 | 0.54 | 0.50 | 0.58 |
| **Ours** | **0.97** | **0.65** | **0.59** | **0.86** |

### 4.3. Subtask 2: Atomic Fact Decomposition

**Dataset.** We collate a human-written atomic fact dataset of 635 examples, each consisting of a paired sentence and list of corresponding human-written atomic facts.

**Methods.** We design the prompt following prior work (Min et al., 2023; Wei et al., 2024), providing in-context examples to illustrate the desired output format (See Appendix E.1). Our task of atomic fact decomposition must account for decontextualization and attribution alignment. We investigate prompting strategies at different levels of granularity: *sentence-level*, in which atomic facts are generated one sentence at a time, versus *response-level*, in which the model generates all atomic facts for the entire response in a single pass. Furthermore, we ablate the decontextualization step, testing the presence or absence of explicit decontextualization, as well as integrating it into a single-pass generation versus treating it as a distinct intermediate step.

**Metric.** To evaluate the similarity between model-generated atomic facts and references, we adopt the metric proposed by Liu et al. (2023b), using Rouge (Lin, 2004) scores calculated via greedy matching. Precision is calculated for each model-generated fact by finding the maximum Rouge-1 F1 score over reference atomic facts and averaging the results. Recall is computed similarly using the reference facts against the generated facts. The final F1 score is the harmonic mean of these precision and recall values. For the F1 score, we strip citations during this phase to focus exclusively on decomposition quality. We also check whether citations are correctly propagated to the corresponding atomic facts (*citation propagation*). Specifically, for each atomic fact derived from a sentence, we consider the match to be correct only if the citation list of the atomic fact is identical to that of the original sentence, with no missing or additional citations. More details can be found in Appendix B.2.

**Results.** As shown in Table 3, the sentence-level approach consistently achieves higher scores compared to response-

level methods. We observe a performance drop when moving to response-level generation, suggesting that prompting models in smaller chunks is crucial for performance. Furthermore, omitting the decontextualization step hurts performance across both sentence and response levels, and asking the model to perform decontextualization implicitly (internally) yields worse results than explicit steps. This confirms the necessity of breaking this complex problem into subtasks. The best performing configuration – explicit decontextualization followed by atomic fact decomposition at the sentence level using Gemini-3-Pro – achieves an F1 of 81.8. Regarding citation accuracy, while Gemini-3-Pro reaches 86.4%, the general trend indicates that correct citation prediction remains a challenging task.

### 4.4. Subtask 3: Attribution Quality

**Dataset.** For the entailment task, we use the atomic facts from verifiable sentences in the human annotations. To evaluate recall and precision, we query the model to provide judgments on combined sources (for recall) and individual sources (for precision) for all verifiable examples. This process yields 917 test examples and 129 validation examples through human annotation.

**Methods & Metric.** We employ the same setup as for verifiable claim identification, but focus on the entailment objective. We adapt the prompt from Jacovi et al. (2025) and utilize the same evaluation metrics (F1 and BAcc).

**Results.** Table 2 shows that the JSON prompt with Gemini-3-Pro achieves the highest F1 (73.1). However, Gemini-2.5-Flash with the Simple prompt is highly competitive, achieving an F1 of 72.9, only 0.2 points behind the best model. Given this marginal difference, we select Gemini-2.5-Flash as the default model for running entailment in our pipeline to maximize efficiency.

### 4.5. End-to-End Evaluation

Finally, we evaluate the metric end-to-end by calculating correlations with human annotation scores. Based on the results in Table 2 and Table 3, our final MURGAT-SCORE employs Gemini-3-Flash for decomposition, Gemini-3-Pro for determining verifiability, and Gemini-2.5-Flash for attri-

bution entailment, balancing performance with cost.

For comparison, we evaluate MURGAT-SCORE against several prompting-based "LLM-as-a-judge" metrics, ranging from response-level judgments to sentence-level granularity. Specifically, we compare against:(1) *Holistic*, which provides a single score ranging from 1–5; (2) *Disentangled*, which asks the model to provide distinct scores for coverage, attribution recall, and precision; and (3) *Disentangled (sentence-level)*, which asks the model to provide these three scores at the sentence level. To ensure strong performance, we use Gemini-3-Pro for these baselines.

Table 4 shows the correlation between human judgments and different evaluation methods. We observe that as we increase in granularity, performance improves; prompting at the sentence level yields notably higher correlations than response-level approaches, particularly for coverage ($r = 0.76$ vs. $0.58$). **MURGAT-SCORE consistently outperforms all baselines across all dimensions, achieving near-perfect correlation on coverage ($r = 0.97$) and strong gains in attribution precision and recall.** This validates the effectiveness of our fine-grained atomic fact decomposition over standard sentence-level prompting. Full correlation results can be seen in Table 16.

# 5. Generation Experiments

Experiments on the human annotation dataset (Table 1) show that even strong MLLMs find MURGAT challenging. In this section, we use our automated evaluation pipeline to investigate why models struggle with this task and to identify factors that improve performance at scale. Section 5.1 describes the experimental setup. Section 5.2 presents results across various base models and citation variants (intrinsic citation generation vs. post-hoc attribution). Finally, we analyze the impact of factors known to improve attribution and reasoning, including programmatic multimodal grounding (Section 5.4) and test-time compute scaling (Section 5.3).

## 5.1. Experimental Setup

We evaluate on Video-MMMU and WorldSense, sampling 100 examples distinct from the human annotation set. We measure answer accuracy via string matching against the gold answer choice (Hong et al., 2025), and MURGAT-SCORE using automatic evaluation. Given our focus on combined audio and visual inputs, we evaluate five representative models capable of handling both modalities: Gemini-2.5-Flash, Gemini-3-Flash, Gemini-3-Pro, Qwen3-Omni-Instruct, and Qwen3-Omni-Thinking. We also include vision-language models that can only process vision information but not audio: Qwen3-VL-instruct, Qwen3-VL-thinking (Bai et al., 2025), and Molmo2-8B (Clark et al., 2026). We evaluate over three variants: (1) direct

generation, where the model provides reasoning and an answer (BASE), (2) generation with citations (+CITATION) following Gao et al. (2023b), and (3) a post-hoc attribution method (POST-HOC ATTRIBUTION), which simulates temporal visual grounding by prompting the model to provide citations for each sentence if necessary. Prompts are shown in Appendix E.

## 5.2. Main Results

We present the primary evaluation in Table 5. Overall, models struggle significantly with multimodal attribution, achieving a peak MURGAT-S of 69.2 on WorldSense and 56.9 on Video-MMMU (Gemini-3-Flash). While Coverage is generally high, attribution remains the bottleneck. Even the best-performing models fail to ground roughly 30-35% of their claims, highlighting the difficulty of precise temporal grounding.

**Impact of Citations is Task-Dependent.** Contrary to the hypothesis that citing evidence always improves performance, we observe a divergence based on task type. On the recognition-focused WorldSense, requiring citations often imposes a "reasoning tax," slightly decreasing accuracy (e.g., Gemini-3-Pro drops from 71.4% to 70.0%), as observed in Zhang et al. (2025); Wan et al. (2025). Conversely, on the reasoning-intensive Video-MMMU, citations often *improve* accuracy (e.g., Gemini-3-Pro improves from 85.3% to 86.0%, and Qwen3-VL-Thinking jumps from 51.0% to 60.0%). This suggests that while citation generation overhead hinders simple retrieval, it may scaffold complex reasoning chains. More details are in Appendix D.2. Models with Chain-of-Thought capabilities (e.g., Qwen3-Omni-Thinking) exhibit a unique failure mode: citations significantly boost their accuracy (e.g., +9.0% on Video-MMMU), yet they struggle to output valid timestamp formats during generation. This results in extremely low citation MURGAT-S scores (e.g., 4.8), requiring Post-hoc methods to recover grounding performance. We further analyze this with program-aided generation in Section 5.4.

**Higher Accuracy ≠ Better Grounding.** Similar to the initial observation in Table 1, high-performing models are not necessarily trustworthy. On Video-MMMU, Gemini-3-Pro (+CITATION) achieves matched accuracy (86.0) with Gemini-3-Flash (+CITATION), yet Gemini-3-Flash maintains a significantly higher MURGAT-S (56.9 vs 41.8). This indicates that stronger models often rely on parametric knowledge to answer correctly while hallucinating supporting citations, underscoring the necessity of MURGAT-S as an independent measure.

**Post-hoc Attribution: Recognition vs. Reasoning.** Applying +POST-HOC ATTRIBUTION yields the highest Coverage, but its impact on attribution quality splits by domain. On WorldSense (recognition), Post-hoc consistently

*Table 5.* Overall performance on WorldSense and Video-MMMU. We report Coverage, Attribution, MURGAT-SCORE (MURGAT-S), and answer accuracy for different model variants. The BASE model does not generate citations; therefore, coverage, attribution, and MURGAT-S are not applicable and left blank. Best results within each method are shown in **bold**.

| Model | Method | WorldSense | | | | Video-MMMU | | | |
|---|---|---|---|---|---|---|---|---|---|
| | | Coverage | Attribution | MURGAT-S | Acc | Coverage | Attribution | MURGAT-S | Acc |
| Gemini-2.5-Flash | BASE | - | - | - | 62.3 | - | - | - | 84.2 |
| | + CITATION | 81.2 | **65.4** | 54.1 | **66.5** | 63.0 | **63.4** | **41.5** | **84.9** |
| | + POST-HOC ATTRIBUTION | **97.4** | 62.3 | **60.8** | 62.3 | **73.8** | 44.9 | 38.0 | 84.2 |
| Gemini-3-Flash | BASE | - | - | - | **67.0** | - | - | - | **86.8** |
| | + CITATION | **95.9** | 66.5 | 64.4 | 66.2 | **88.2** | **64.5** | **56.9** | 86.0 |
| | + POST-HOC ATTRIBUTION | 95.1 | **71.4** | **69.2** | **67.0** | 87.9 | 47.2 | 44.1 | **86.8** |
| Gemini-3-Pro | BASE | - | - | - | **71.4** | - | - | - | 85.3 |
| | + CITATION | 78.3 | 64.9 | 51.7 | 70.0 | 63.4 | **67.3** | **41.8** | **86.0** |
| | + POST-HOC ATTRIBUTION | **97.0** | **67.1** | **65.2** | **71.4** | **68.0** | 43.7 | 36.9 | 85.3 |
| Qwen3-Omni-Instruct | BASE | - | - | - | **57.0** | - | - | - | **45.0** |
| | + CITATION | 47.6 | **53.3** | 29.0 | 54.0 | 34.6 | **21.8** | 9.8 | 40.0 |
| | + POST-HOC ATTRIBUTION | **99.5** | 45.7 | **45.4** | **57.0** | **95.1** | 17.9 | **17.6** | **45.0** |
| Qwen3-Omni-Thinking | BASE | - | - | - | 56.5 | - | - | - | **53.0** |
| | + CITATION | 52.7 | 56.3 | 31.3 | **61.0** | 36.3 | 7.6 | 4.8 | 51.0 |
| | + POST-HOC ATTRIBUTION | **93.2** | **60.0** | **56.3** | 56.5 | **76.3** | **16.8** | **12.8** | **53.0** |
| *Vision-Language Only* | | | | | | | | | |
| Qwen3-VL-Instruct | BASE | - | - | - | **50.0** | - | - | - | 53.0 |
| | + CITATION | 39.0 | 52.0 | 25.5 | 48.0 | 30.2 | 40.1 | 17.5 | **55.0** |
| | + POST-HOC ATTRIBUTION | **98.9** | **70.2** | **69.4** | **50.0** | **93.4** | **44.6** | **42.3** | 53.0 |
| Qwen3-VL-Thinking | BASE | - | - | - | 47.0 | - | - | - | 51.0 |
| | + CITATION | 38.5 | 56.1 | 30.8 | **49.0** | 23.2 | 15.1 | 7.6 | **60.0** |
| | + POST-HOC ATTRIBUTION | **76.6** | **58.9** | **48.2** | 47.0 | **54.3** | **31.5** | **18.9** | 51.0 |
| Molmo2 | BASE | - | - | - | 41.0 | - | - | - | 50.5 |
| | + CITATION | 69.1 | **50.2** | **39.7** | 40.0 | **82.6** | **21.4** | **19.3** | 44.3 |
| | + POST-HOC ATTRIBUTION | **75.0** | 38.3 | 33.2 | **41.0** | 66.4 | 15.0 | 11.4 | **50.5** |

improves MURGAT-S (e.g., Gemini-3-Pro: 51.7 → 65.2) by accurately locating visual entities. However, on Video-MMMU (reasoning), Post-hoc causes Attribution to plummet (e.g., Gemini-2.5-Flash: 41.5 → 38.0). Qualitatively, post-hoc methods tend to "force-align" abstract reasoning steps to random segments, creating false positives. More details are discussed in Appendix D.2.

**Omni Models vs. Vision-Language Baselines.** We observe a distinct trade-off between modality breadth and reasoning depth. On WorldSense, Vision-Language (VL) models achieve low accuracy due to the lack of audio processing; consequently, Qwen3-Omni-Instruct significantly outperforms Qwen3-VL-Instruct (57.0% vs 50.0%). This trend reverses on the reasoning-intensive Video-MMMU (53.0% vs 45.0%), likely because VL models prioritize long-context visual encoding, avoiding the real-time streaming trade-offs inherent to Omni architectures. However, comparable attribution scores between these model families can be misleading. As detailed in Table 17, VL models frequently hallucinate audio citations—comprising up to 31.6% of their references despite lacking an audio encoder. This indicates that their "grounding" often relies on visual proxies or hallucinations rather than genuine auditory understanding. Thus, a high MURGAT-S for VL models merely reflects an ability to ground *observations*, which are often irrelevant visual

details rather than the causal reasoning chain required to reach the gold answer.

### 5.3. Impact of Reasoning Effort

While increased reasoning depth typically improves task performance, its impact on attribution is less clear. We analyze models across different "thinking" effort levels (Minimal to High). As shown in Figure 2, we observe diverging trends between models. For Gemini-3-Flash on WorldSense, increased reasoning effort counter-intuitively leads to a decline in attribution quality, with MURGAT-SCORE dropping from 69.7 (Minimal) to 64.4 (High). This suggests that for the Flash model, internal latent reasoning may be somewhat incompatible with the explicit retrieval required for external verification. Interestingly, on Video-MMMU, Gemini-3-Flash peaks at **Medium** effort (91.5% Accuracy), indicating a specific "sweet spot" for reasoning duration.

In contrast, Gemini-3-Pro demonstrates positive scaling on WorldSense: increasing reasoning effort from Low to High results in a +6.1 point increase in MURGAT-SCORE and a +7.4 point boost in accuracy. This indicates that stronger models are better equipped to align their reasoning chains with external evidence.

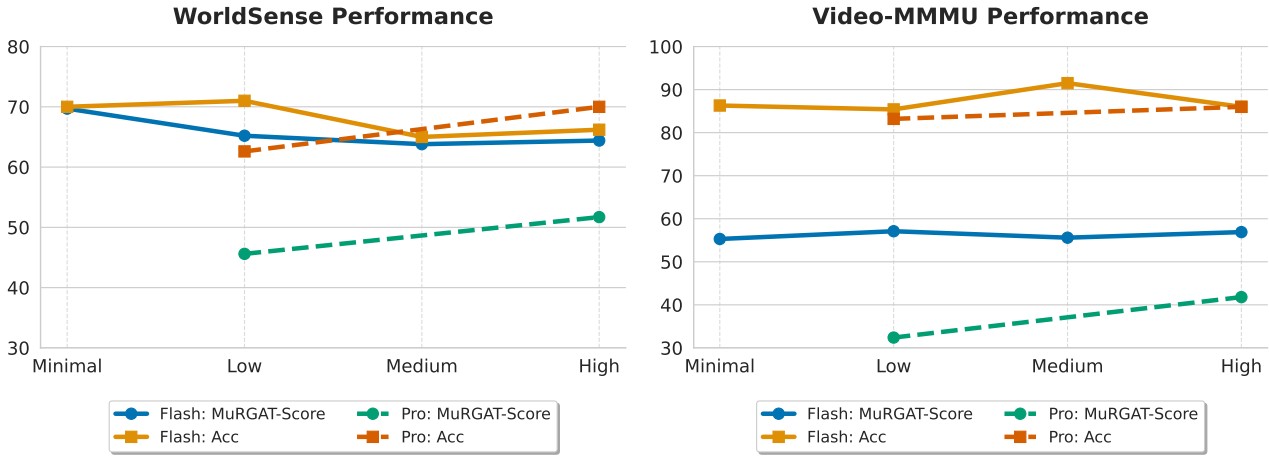

*Figure 2.* Gemini models' performance on MᴜRGᴀᴛ with different thinking-effort levels (Minimal, Low, Medium, High) under the +Cɪᴛᴀᴛɪᴏɴ setting from Table 5. The "High" setting corresponds to the default +Cɪᴛᴀᴛɪᴏɴ configuration reported elsewhere.

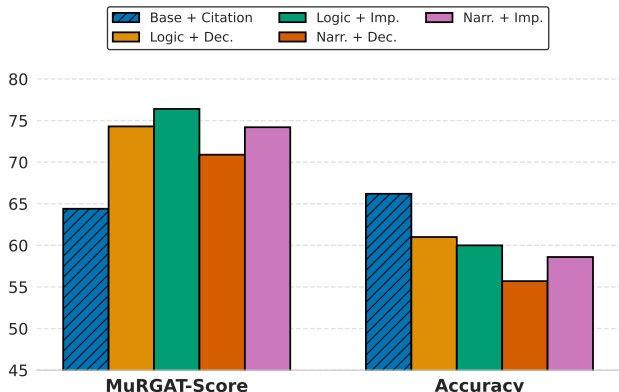

*Figure 3.* Gemini-3-Flash results with program-aided generation on WorldSense.

### 5.4. Programmatic Multimodal Grounding

To evaluate how frameworks designed to improve attribution quality perform on our benchmark, we extend prior work on program-aided generation (Wan et al., 2025; Slobodkin et al., 2024) to our challenging multimodal setting along two axes: (1) **Reasoning Paradigm**: *Logic-Centric* (imperative Python-like code) vs. *Narrative-Centric* (declarative natural language steps); and (2) **Grounding Mechanism**: *Declarative* (Planner-Defined), where the model predicts timestamps directly, vs. *Imperative* (Executor-Discovered), where the model generates search queries for a retrieval tool. Complementing these axes, we integrate a runtime refinement mechanism to verify that atomic operations are strictly entailed by input evidence, ensuring high grounding fidelity throughout the execution loop.

As shown in Figure 3 and Table 14, program-aided frameworks consistently enhance attribution quality on World-sense. Compared to the Bᴀsᴇ + Cɪᴛᴀᴛɪᴏɴ baseline, programmatic methods yield an average MᴜRGᴀᴛ-Sᴄᴏʀᴇ

gain of +9.6 points, with the Lᴏɢɪᴄ Iᴍᴘᴇʀᴀᴛɪᴠᴇ variant achieving the highest performance (76.4). Notably, **Imperative** methods consistently outperform **Declarative** ones (e.g., Logic Imperative 76.4 vs. Declarative 74.3), suggesting that allowing models to execute search queries is more effective than direct timestamp prediction.

However, this improvement in attribution comes at the cost of answer accuracy, which drops by an average of 7.4 points. This trade-off aligns with observations by Wan et al. (2025), suggesting that while explicit structuring aids verification, it may constrain the model's inherent reasoning flexibility.

## 6. Conclusion

We introduced MᴜRGᴀᴛ, a benchmark designed to evaluate fact-level attribution in multimodal large language models. Unlike prior tasks focused on retrieval or simple observation, MᴜRGᴀᴛ targets complex scenarios requiring models to synthesize answers from video, audio, and figures while providing precise evidentiary support. To evaluate this rigorously, we developed MᴜRGᴀᴛ-Sᴄᴏʀᴇ, a decomposed, fine-grained automatic evaluation pipeline with high correlation to human judgments. Our extensive experiments with state-of-the-art MLLMs reveal that the capability to reason does not imply the capability to ground. We identified key failure modes, including the tendency of post-hoc methods to hallucinate mappings in complex reasoning tasks and the trade-off between programmatic rigor and narrative accuracy. We hope MᴜRGᴀᴛ and MᴜRGᴀᴛ-Sᴄᴏʀᴇ facilitate future research into reconciling these capabilities, moving towards MLLMs that are both accurate and faithful.

## Acknowledgments

We would like to thank the annotators: Nithin Sivakumaran, Tianyi Niu, Atharv Sumant Kulkarni, Fengli Wu, and Salvador Robles Herrera. This work was supported by ONR Grant N00014-23-1-2356, ARO Award W911NF2110220, DARPA ECOLE Program No. HR00112390060, NSF-AI Engage Institute DRL2112635, NSF-CAREER Award 1846185, Microsoft Accelerating AI Academic Research (AARI) program, and a Google PhD Fellowship. The views contained in this article are those of the authors and not of the funding agency.

## Impact Statement

While our automatic evaluation pipeline correlates highly with human judgments, it relies on model-based verification, which may carry inherent biases or limitations that should be considered before deploying it as a sole arbiter of truth in sensitive contexts.

In our work, we have designed the evaluation protocol to strictly filter for verifiable claims and rely on granular atomic facts to mitigate ambiguity and bias. We do not believe our method introduces new risks beyond those inherent in existing multimodal generation technologies. Conversely, by establishing a rigorous standard for citation and evidence, we aim to provide the research community with the necessary tools to detect errors, reduce hallucination, and build more trustworthy systems.

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

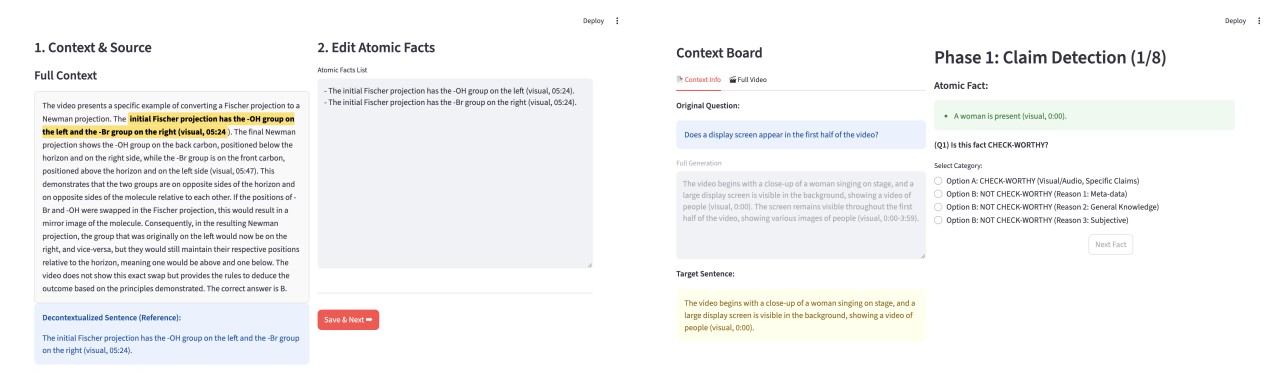

*(a)* Annotation UI for Atomic Fact Decomposition.    *(b)* Annotation UI for Verification Worthiness.

# A. Human Evaluation Details

To validate our automatic metrics, we developed a multi-stage human annotation protocol. The protocol consists of three stages: atomic decomposition, verifiable claim identification, and attribution quality. While the evaluation protocol described in Section 3.2 performs the verifiable claim identification at the sentence-level, we ask annotators to do this step at an atomic-fact level for comprehensiveness that allows future research with more fine-grained data. Note that in our evaluation framework, we use sentence-level verifiable claim identification, as we observe similar performance but much lower cost, as detailed in Appendix B.1.

## A.1. Data and Models

To encompass diverse model behaviors, we sampled inputs from Video-MMMU (Hu et al., 2025b), which focuses on figures and graphs with audio, and WorldSense (Hong et al., 2025), which emphasizes video and audio interpretation. As these require models to have both visual and audio reasoning capabilities, we evaluated four MLLMs: *Gemini-2.5-Flash (Comanici et al., 2025)*, *Gemini-3-Pro (Google, 2025)*, *Qwen3-Omni-Instruct*, and *Qwen3-Omni-Thinking* (Xu et al., 2025). The models were prompted to generate answers containing reasoning processes and citations. See prompt in Figure 11. We randomly select 10 examples from Video-MMMU and WorldSense each, resulting in 80 generations. These 80 generations yielded a total of 600 sentences.

## A.2. Atomic Decomposition

**Guidelines.**   Two annotators decomposed complex sentences into independent atomic units according to the following guidelines: Pronouns were resolved using strictly prior context (forward-only) to prevent information leakage, while meta-talk (e.g., *"The video shows"*) was stripped. A critical addition to our protocol is *manual citation propagation*. Rather than inheriting all citations from the source sentence, annotators assigned specific timestamps (e.g., distinct visual vs. audio evidence) strictly to their relevant atomic facts. Finally, logical reasoning steps, mathematical operations, and compound visual attributes were decomposed to allow for precise partial-credit verification. The annotation interface is shown in Figure 4a.

**Annotation Details.**   To accelerate the process, we used Gemini-3-Pro to generate an initial candidate list of facts, similar to Min et al. (2023). Following this drafting phase, annotators manually refined the outputs. This included decontextualization, where annotators resolved pronouns and ambiguous references based on the full generation context to ensure each fact was self-contained. The process required an average of 35.7 seconds per sentence, with consensus resolution taking an additional 48.2 seconds. On average, the dataset contains 25.6 atomic facts per response.

## A.3. Verifiable Claim Identification Annotation

**Guidelines.**   Annotators evaluated each atomic fact to determine if it describes verifiable video content, a process referred to as verifiable claim identification. A fact is classified as verifiable if it describes specific visual or audio events, claims such as dates and locations, or the absence of an object. Conversely, facts are filtered out as non-verifiable based on three criteria: *Task Meta-data & Reasoning* (e.g., "Therefore, Option A is correct."), *General Knowledge & Definitions* (e.g., "Cars are

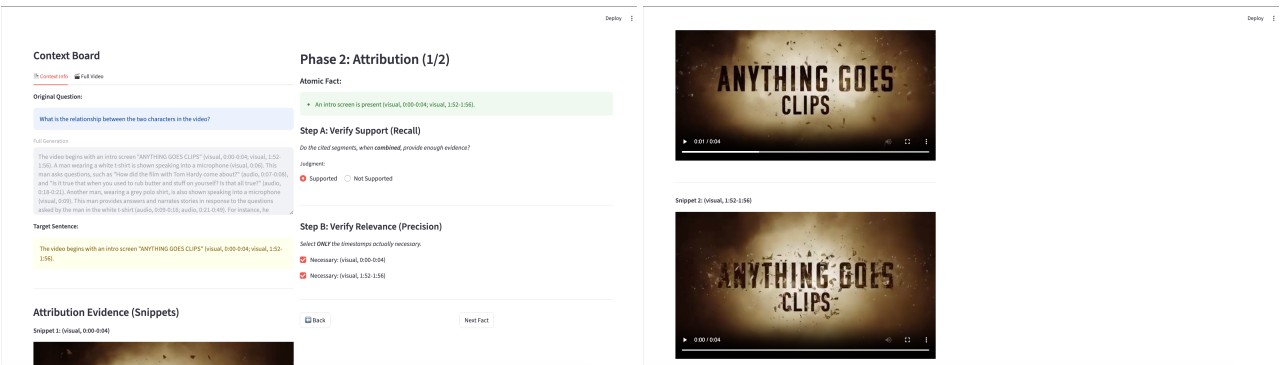

*Figure 5.* Annotation UI for Attribution.

vehicles."), or *Subjective / Chitchat* (e.g., "I hope this helps."). This judgment was performed strictly at the atomic-fact level to ensure granular coverage. The annotation interface is illustrated in Figure 4b.

**Annotation Details.** During this stage, we observed a moderate inter-annotator agreement of 73.7%. Analysis revealed that these disagreements were primarily due to varying sensitivity thresholds, where one annotator might miss a subtle verifiable claim, as evidenced by the significantly higher agreement in the subsequent attribution evaluation stage. Consequently, we adopted a Union Strategy (OR-gate) for this phase, retaining any atomic fact marked as verifiable by at least one annotator. This inclusive approach preserved 15.2% of the dataset (N=216) that would have been discarded under a strict consensus model, thereby ensuring high recall. While our final proposed evaluation framework utilizes a simplified sentence-level verifiable claim identification followed by atomic decomposition, this atomic-level annotation was essential for establishing a high-quality, high-recall gold standard.

**Over-citation Analysis.** Although our primary coverage metric focuses on verifiable facts, we also investigated instances where the model provides citations for sentences deemed not verifiable (over-citation). Our analysis identified 37 sentences classified as not verifiable by humans; of these, only 3 sentences contained model citations. This represents an over-citation rate of only 8%, suggesting that while over-citation occurs, it affects only a small portion of the non-verifiable content.

### A.4. Attribution Annotation

**Guidelines.** Human annotators validated the model-generated timestamps for facts deemed verifiable in the previous phase. To maximize efficiency and ensure the context of the atomic fact remained fresh in the annotator's mind, we combined the verification-worthiness and attribution tasks into a single annotation run. Once a fact was marked as verifiable, the interface immediately prompted the annotator to evaluate the attribution across two dimensions. First, they evaluated *Recall (Support)*, determining if the cited segments, when combined, provided sufficient evidence to entail the fact. Second, if the fact was supported, they evaluated *Precision (Necessity)* by selecting the specific checkboxes for only those timestamps strictly required to prove the claim. This second step allowed annotators to filter out irrelevant timestamps, effectively penalizing "citation dumping" behaviors. The annotation UI is shown in Figure 5.

**Annotation Details.** The inter-annotator agreement on the verification of these claims reached 86.1%. This level of reliability is notably high and compares favorably to similar state-of-the-art verification benchmarks, such as Liu et al. (2023a), which reported a pairwise agreement of 82.2%. This strong consensus justifies our use of the Union Strategy in the preceding phase, as it confirms that annotators are highly consistent once a claim has been identified for checking.

### A.5. Handling Multi-Source Sentences

A natural question with our protocol is whether sentences carrying citations from multiple modalities are unfairly penalized when only a subset of the citations is needed for each atomic fact. The protocol evaluates each citation against the specific atomic fact it supports rather than penalizing all facts collectively. As a concrete example, the Figure 1 sentence *"The video explicitly defines the convention that 'repulsive forces are positive' on the graph (audio, 0:42–0:46; visual, 0:45)"* decomposes into two atomic facts: (F1) *repulsive forces are positive*, and (F2) *a convention is defined*. The citation set $C_i = \{(\text{audio}, 0:42–0:46), (\text{visual}, 0:45)\}$ propagates to both facts, but each citation is verified independently per fact. F1 is fully entailed by the union of both segments (recall: supported); F2 is entailed by the audio segment alone, so the visual

*Table 6.* Citation placement patterns across annotated outputs. End-dumped citations are the dominant pattern across all model families.

| Model | Total | Multi-cite | Inline | End-dumped | % End-dumped |
|---|---|---|---|---|---|
| Gemini-2.5-Flash | 225 | 58 (25.8%) | 11 | 47 | 81.0% |
| Gemini-3-Pro | 127 | 21 (16.5%) | 6 | 15 | 71.4% |
| Qwen3-Omni-Instruct | 136 | 5 (3.7%) | 0 | 5 | 100.0% |
| Qwen3-Omni-Thinking | 112 | 5 (4.5%) | 1 | 4 | 80.0% |
| **All** | **600** | **89 (14.8%)** | **18** | **71** | **79.8%** |

*Table 7.* Summary of Annotation Statistics and Model Performance. $N_{sent}$ and $N_{fact}$ denote the total number of sentences and facts evaluated per model. Coverage, Attribution Recall, Precision, F1, MURGAT-S, and Accuracy are reported as percentages.

| Model | WorldSense | | | | | | | | Video-MMMU | | | | | | | |
|---|---|---|---|---|---|---|---|---|---|---|---|---|---|---|---|---|
| | $N_{sent}$ | $N_{fact}$ | Cov. | Rec. | Prec. | F1 | MURGAT-S | Acc. | $N_{sent}$ | $N_{fact}$ | Cov. | Rec. | Prec. | F1 | MURGAT-S | Acc. |
| Qwen3-Omni-Instruct | 43 | 139 | 55.1 | 35.4 | 49.1 | 41.1 | 27.3 | 56.0 | 93 | 282 | 35.9 | 14.9 | 49.1 | 22.9 | 5.6 | 67.4 |
| Qwen3-Omni-Thinking | 35 | 151 | 47.1 | 41.2 | 67.4 | 51.1 | 23.1 | 56.0 | 77 | 309 | 45.0 | 23.4 | 67.4 | 34.7 | 21.8 | 76.0 |
| Gemini-2.5-Flash | 72 | 237 | 85.0 | 65.8 | 59.7 | 62.6 | 59.9 | 58.0 | 153 | 482 | 57.1 | 32.5 | 59.7 | 42.1 | 21.8 | 72.0 |
| Gemini-3-Pro | 40 | 146 | 76.8 | 61.6 | 57.8 | 59.6 | 49.7 | 60.0 | 87 | 299 | 59.7 | 24.8 | 57.8 | 34.7 | 16.3 | 86.0 |

segment is flagged as unnecessary for F2 (precision: 0.5 for F2, 1.0 for F1). The result is sentence-level recall $= 1.0$ and precision $= 0.75$, not a blanket failure. This per-fact verification mirrors text attribution protocols (Gao et al., 2023b; Liu et al., 2023a).

## A.6. Citation Placement Analysis

To assess whether attaching the full citation set $C_i$ to every atomic fact derived from sentence $r_i$ unfairly penalizes sub-facts when only a subset of citations is relevant to each, we analyze citation placement patterns across all 600 annotated sentences and check how often models embed citations *inline* (next to the specific claim they support) versus *end-dumped* (clustered at the end of the sentence). We classify sentences automatically by detecting whether multiple citations are interleaved with sentence text or grouped at the boundary.

Table 6 reports the breakdown. Only 14.8% of sentences contain multiple citations, and within those, 79.8% follow an end-dumped pattern while 20.2% are inline. Two findings justify the propagation rule. First, our decomposition prompt (Figure 9) includes a **Splitting rule** that assigns inline citations only to the relevant atomic fact, correctly handling the 20.2% inline cases. Second, examining the 71 end-dumped sentences in our human annotations, annotators assigned the full citation set to every atomic fact in 100% of cases, judging each citation as applicable to all facts rather than a subset. This indicates that when models cluster citations at the end, the cited segments tend to span the full sentence content, making blanket propagation an accurate reflection of the model's intent. The effect is also symmetric across models (71–100% end-dumping rate), so rankings are unaffected, and recall is entirely unaffected since recall pools the union of citations.

## A.7. Full Statistics

We show the full statistics of Table 1 in Table 7, showing the number of sentences, number of atomic facts, and also the detailed breakdown of attribution recall and precision.

# B. Automatic Evaluation Details

## B.1. Verifiable Claim Identification

To evaluate verifiable claim identification, we adapt human annotations by treating verifiable claims as positive instances and all other claims as negative instances. Given the text-centric nature of this task, we expand our evaluation beyond the Gemini family to include Gemma-3-27b-it (Gemma Team, 2025) and GPT-5.2 (OpenAI, 2025), with results for both sentence-level and atomic-fact level granularity presented in Table 8. Gemini-3-Pro achieves the highest performance across both levels, followed closely by GPT-5.2 (CoT) at the sentence level by a narrow 0.3-point margin. While performance trends remain consistent across granularities, we observe that Balanced Accuracy (BAcc) scores are highly comparable. Consequently, due to the significantly higher computational cost associated with atomic-fact decomposition, we adopt sentence-level evaluation as our primary metric for the remainder of this study.

*Table 8.* Verifiable Claim Identification results comparing Sentence-level and Atomic-fact level performance (BAcc).

| | | Balanced Accuracy (BAcc) | |
|---|---|---|---|
| Model | Method | Sentence-level | Atomic-fact level |
| Gemini-2.5-Flash | Simple | 78.0 | 68.2 |
| | CoT | 75.8 | 71.6 |
| | JSON | 80.6 | 73.7 |
| Gemini-3-Flash | Simple | 80.8 | 78.2 |
| | CoT | 80.2 | 75.3 |
| | JSON | 81.1 | 77.0 |
| Gemini-3-Pro | Simple | 79.0 | **81.7** |
| | CoT | 81.4 | 74.3 |
| | JSON | **84.2** | 80.8 |
| Gemma-3-27b-it | Simple | 79.8 | 68.8 |
| | CoT | 68.8 | 67.7 |
| | JSON | 76.0 | 73.8 |
| GPT-5.2 | Simple | 81.3 | 75.0 |
| | CoT | 83.9 | 72.7 |
| | JSON | 80.7 | 75.0 |

*Table 9.* Full correlation results for atomic fact decomposition. We compare the full (*Full*) pipeline against ablations without decontextualization (*w/o Decontext.*) and a combined single-pass generation (*Single Pass*).

| Model | Format | Sentence-level | | Response-level | |
|---|---|---|---|---|---|
| | | F1 | Cit. Acc. | F1 | Cit. Acc. |
| Gemini-2.5-Flash | Full | 81.0 | 85.5 | 77.8 | 79.9 |
| | w/o Decontext. | 78.7 | 84.2 | 77.3 | 78.2 |
| | Single Pass | 77.5 | 81.6 | 78.4 | 80.5 |
| Gemini-3-Flash | Full | 81.4 | 85.3 | 79.7 | 81.4 |
| | w/o Decontext. | 79.0 | 84.0 | 78.5 | 81.9 |
| | Single Pass | 77.7 | 82.7 | 77.8 | 80.0 |
| Gemini-3-Pro | Full | **81.8** | **86.4** | **80.1** | **84.7** |
| | w/o Decontext. | 79.8 | 85.2 | 79.0 | 84.0 |
| | Single Pass | 78.8 | 83.9 | 79.7 | 82.7 |
| Gemma-3-27b-it | Full | 79.3 | 74.3 | 74.0 | 66.4 |
| | w/o Decontext. | 77.8 | 71.7 | 74.2 | 66.1 |
| | Single Pass | 78.2 | 63.8 | 73.9 | 60.8 |
| GPT-5.2 | Full | 81.2 | 82.3 | 73.3 | 76.3 |
| | w/o Decontext. | 78.2 | 82.2 | 70.1 | 70.9 |
| | Single Pass | 73.0 | 75.4 | 69.5 | 71.9 |

## B.2. Atomic Fact Decomposition

We present full results for atomic fact decomposition in Table 9, including additional models such as Gemma-3-27b-it and GPT-5.2. We also evaluate a response-level approach, where the model generates all atomic facts for the entire response in a single pass. As shown in the table, performance drops noticeably at the response level compared to the sentence level.

Our results underscore the importance of explicit decontextualization and the separation of the pipeline into distinct stages. For example, Gemini-3-Pro achieves a 2-point gain in F1 when using decontextualization compared to when it is omitted. Furthermore, separating decontextualization and decomposition into two stages yields a 3-point gain over the single-pass method (where the model performs both implicitly). This confirms the utility of a two-stage pipeline for generating high-quality atomic facts. Similarly, citation accuracy is consistently highest when the process is decomposed into two stages.

*Table 10.* Correlations with human judgments across evaluator configurations. Subtask 3 is held to a multimodal-capable model; Subtasks 1–2 are varied. All correlations are significant at $p < 0.001$.

| Evaluator (Subtasks 1–2 / Subtask 3) | Pearson | Spearman | Kendall |
|---|---|---|---|
| Ours (Gemini-3-Pro, Gemini-3-Flash / Gemini-2.5-Flash) | 0.860 | 0.844 | 0.685 |
| Gemini-3-Pro only | 0.879 | 0.853 | 0.697 |
| Gemini-2.5-Flash only | 0.830 | 0.836 | 0.679 |
| GPT-5.2 / Gemini-2.5-Flash | 0.757 | 0.776 | 0.609 |

*Table 11.* Bootstrap 95% CIs (10,000 resamples) for Spearman correlation with human judgments. All correlations significant at $p < 0.001$.

| Evaluator | Metric | Spearman | 95% CI |
|---|---|---|---|
| Ours | MURGAT-SCORE | 0.844 | [0.737, 0.911] |
| Ours | Coverage | 0.967 | [0.939, 0.983] |
| GPT-5 / Gemini-2.5-Flash | MURGAT-SCORE | 0.776 | [0.641, 0.868] |
| Gemini-2.5-Flash only | MURGAT-SCORE | 0.836 | [0.722, 0.910] |
| Gemini-3-Pro only | MURGAT-SCORE | 0.853 | [0.744, 0.921] |

*Table 12.* Bootstrap subsampling stability of MURGAT-SCORE Spearman correlation (10,000 resamples per size).

| Sample Size | Median Spearman | 95% CI | CI Width |
|---|---|---|---|
| 30 | 0.841 | [0.642, 0.940] | 0.298 |
| 50 | 0.842 | [0.700, 0.924] | 0.223 |
| 80 | 0.843 | [0.737, 0.911] | 0.174 |

## B.3. Robustness to Evaluator Choice

We analyze whether Gemini models as evaluators biases scores toward Gemini-generated outputs. Our pipeline already uses different models for different subtasks (Gemini-3-Pro, Gemini-3-Flash, and Gemini-2.5-Flash), so no single model evaluates its own outputs end-to-end. We further test alternative evaluator configurations in which Subtasks 1–2 are replaced with a single model. Subtask 3 (entailment) remains Gemini-2.5-Flash or Gemini-3-Pro because it requires multimodal input – GPT does not natively process video/audio streams, and Qwen evaluators achieve substantially lower correlation (best F1: 57.8 / 58.7 for Qwen3-Omni-Instruct / Qwen3-Omni-Thinking vs. 73.1 for Gemini-3-Pro).

Table 10 reports correlations with human judgments across configurations (all $p < 0.001$). All variants achieve strong correlations, with our multi-model pipeline performing comparably to single-evaluator alternatives. To check for in-family inflation, we examined how each evaluator scores Gemini outputs relative to human rankings: our pipeline achieves **perfect rank-order agreement** ($\tau = 1.00$) with human rankings, while a GPT-5 evaluator inverts the top two model rankings ($\tau = 0.33$). The Gemini-3-Pro evaluator scores its own model's outputs (40.0) *lower* than Gemini-2.5-Flash outputs (44.7), and our metric's spread (23.4) closely tracks the human-judged spread (24.4), indicating no systematic in-family bias.

## B.4. Statistical Reliability of Correlations

While our human evaluation set comprises 20 inputs, the multi-model and multi-stage design produces 80 model responses, 580 verifiability labels, 635 decomposition annotations, and 917 entailment labels, which is comparable in scale to prior attribution validation sets (Jacovi et al., 2025). To verify that correlations are not artifacts of small-sample variance, we conduct bootstrap significance tests (10,000 resamples) for each evaluator configuration. Table 11 reports 95% confidence intervals for Spearman correlation; all coefficients are significant at $p < 0.001$ with tight intervals.

We also conduct bootstrap subsampling at smaller annotation sizes to assess sensitivity (Table 12). The median Spearman remains $\approx 0.84$ across sample sizes from 30 to 80, with confidence-interval width narrowing monotonically as $n$ grows. This indicates the correlation is not driven by a small subset of outliers, and that the 80-response evaluation set is sufficient for stable rank-level conclusions.

## B.5. Manual Spot-Check of Pipeline Outputs

To complement the human-correlation analysis in Table 4, we manually inspect 50 randomly sampled entailment judgments produced by the automated pipeline. The authors review each citation against the underlying video and audio evidence and independently re-judge entailment. Of the 50 samples, 47 (94%) match the pipeline's judgment; the 3 disagreements involve borderline cases with ambiguous temporal boundaries (e.g., utterances spanning the boundary between two cited segments). This high agreement rate corroborates the human-correlation results and provides additional confidence that the automated judge produces reliable assessments at scale.

# C. Programmatic Multimodal Grounding

We introduce a framework designed to improve grounding fidelity by structurally decoupling reasoning from attribution. Inspired by recent advances in program-aided generation (Wan et al., 2025; Slobodkin et al., 2024), the model operates on a "plan-then-execute" paradigm. Rather than generating a direct textual response, the model first constructs a structured plan composed of executable modules. This approach ensures that every claim is explicitly linked to a retrieved source, allowing for automatic and verifiable citation assignment.

Our primary research objective is to identify the optimal programmatic structure for faithful multimodal grounding. To this end, we explore the design space along two orthogonal axes: the *Reasoning Paradigm* (the style of the program) and the *Grounding Mechanism* (how evidence is localized).

## C.1. Axis 1: Reasoning Paradigm

This axis defines the semantic structure of the generated program and the nature of its intermediate artifacts. We contrast two dominant approaches:

**Logic-Centric.** Exemplified by ViperGPT (Surís et al., 2023), this paradigm treats the multimodal source as a structured database to be queried. The generated programs are imperative (e.g., Python scripts) utilizing control flow (loops, conditionals) and abstract variables (e.g., boolean flags, integer counts). While highly effective for verifiable, objective queries (e.g., *"How many muffins are on the table?"*), the intermediate steps are often opaque data structures that lack human-readable context, potentially obscuring the reasoning chain.

**Narrative-Centric.** Exemplified by Generation Programs (Wan et al., 2025), this paradigm treats the source as a narrative to be reconstructed. The program consists of declarative function calls (e.g., `describe`, `synthesize`) that produce semantic, natural language outputs at every step. This style prioritizes *contributive attribution*, ensuring that the reasoning trace itself serves as a verifiable, human-readable explanation of the final answer.

## C.2. Axis 2: Grounding Mechanism

This axis defines *when* and *how* specific evidentiary segments (timestamps, bounding boxes) are identified within the pipeline. We investigate the trade-off between planner control and executor robustness.

**Planner-Defined (Declarative Grounding).** In this setting, the MLLM perceives the video content during the planning phase and explicitly commits to citations within the generated code (e.g., `describe('00:15-00:20', ...)`). This mimics text-based retrieval approaches where models select sentence indices from a context window. This approach grants the planner maximum control over the narrative flow but relies heavily on the MLLM's internal ability to localize events without hallucination.

**Executor-Discovered (Imperative Grounding).** Here, the MLLM delegates the localization task to a specialized tool during execution (e.g., `events = find('boy holding ball')`). Rather than hypothesizing timestamps, the planner instead defines the *search criteria*. This approach is theoretically more robust against hallucination, as it relies on the recall of the retrieval tool rather than the model's parametric memory, but it shifts the burden of performance to the retrieval tool.

## C.3. Refinement Mechanism

To further enhance grounding fidelity, we integrate a post-hoc optimization strategy into the execution loop. Building on findings that structured programs facilitate verification (Wan et al., 2025), we implement a runtime attribution check, which showed improvement in grounding performance in early experiments. After each execution step, we verify that the output of a function call is entailed by its input evidence. This ensures that individual atomic operations maintain high attribution standards before their results are aggregated into the final response.

## C.4. Implementation

We instantiate the model as a Python-based framework capable of operating across both axes described above. The core library consists of three atomic operations adapted for multimodal inputs:

1. `find_event(query)` → `List[Timestamp]`: A retrieval tool to locate relevant segments based on semantic queries.
2. `describe(timestamp | event_ref, instruction)` → `str`: A vision-language call that inspects a specific segment and generates a dense textual description grounded in the visual evidence.
3. `synthesize(evidence_list, instruction)` → `str`: A logical deduction step that aggregates previous descriptions to answer the user query without accessing the raw video, forcing reliance on the retrieved evidence.

## C.5. Results

This structure imposes a penalty on complex reasoning tasks; on Video-MMMU, the base models consistently outperform the programmatic variants in accuracy (e.g., a drop from 90.0% to 84.7% for Gemini-3-Flash), indicating that while enforcing a "plan-then-execute" structure curbs "correct for the wrong reasons" behavior, it may excessively constrain the model's flexibility on questions requiring holistic video understanding.

*Table 13.* Full results on WorldSense and Video-MMMU. We report Coverage, Attribution (Precision, Recall, and F1), MURGAT-SCORE (MURGAT-S), and answer accuracy for different model variants. Best results within each method are shown in **bold**.

| Model | Method | WorldSense | | | | | | Video-MMMU | | | | | |
|---|---|---|---|---|---|---|---|---|---|---|---|---|---|
| | | Cov. | Attr. P | Attr. R | Attr. F1 | MURGAT-S | Acc | Cov. | Attr. P | Attr. R | Attr. F1 | MURGAT-S | Acc |
| Gemini-2.5-Flash | BASE | - | - | - | - | - | 62.3 | - | - | - | - | - | 84.2 |
| | + CITATION | 81.2 | **64.2** | 67.0 | **65.4** | 54.1 | **66.5** | 63.0 | 59.6 | 68.5 | 63.4 | **41.5** | **84.9** |
| | + POST-HOC ATTRIBUTION | **97.4** | 60.9 | 64.3 | 62.3 | **60.8** | 62.3 | **73.8** | 42.5 | 48.0 | 44.9 | 38.0 | 84.2 |
| Gemini-3-Flash | BASE | - | - | - | - | - | 67.0 | - | - | - | - | - | **86.8** |
| | + CITATION | **95.9** | 64.0 | 69.7 | 66.5 | 64.4 | 66.2 | **88.2** | 59.9 | 71.0 | 64.5 | **56.9** | 86.0 |
| | + POST-HOC ATTRIBUTION | 95.1 | **68.8** | **75.2** | **71.4** | **69.2** | **67.0** | 87.9 | 43.6 | 52.3 | 47.2 | 44.1 | **86.8** |
| Gemini-3-Pro | BASE | - | - | - | - | - | 71.4 | - | - | - | - | - | 85.3 |
| | + CITATION | 78.3 | 63.6 | 66.6 | 64.9 | 51.7 | 70.0 | 63.4 | **64.6** | **71.3** | **67.3** | **41.8** | **86.0** |
| | + POST-HOC ATTRIBUTION | **97.0** | **65.4** | **69.6** | **67.1** | **65.2** | **71.4** | **68.0** | 41.0 | 47.2 | 43.7 | 36.9 | 85.3 |
| Qwen3-Omni-Instruct | BASE | - | - | - | - | - | **57.0** | - | - | - | - | - | **45.0** |
| | + CITATION | 47.6 | **53.2** | **53.7** | **53.3** | 29.0 | 54.0 | 34.6 | **22.0** | **22.8** | **21.8** | 9.8 | 40.0 |
| | + POST-HOC ATTRIBUTION | **99.5** | 45.7 | 46.5 | 45.7 | **45.4** | **57.0** | **95.1** | 17.9 | 17.9 | 17.9 | **17.6** | **45.0** |
| Qwen3-Omni-Thinking | BASE | - | - | - | - | - | 56.5 | - | - | - | - | - | **53.0** |
| | + CITATION | 52.7 | 56.4 | 56.4 | 56.3 | 31.3 | **61.0** | 36.3 | 7.8 | 8.3 | 7.6 | 4.8 | 51.0 |
| | + POST-HOC ATTRIBUTION | **93.2** | **59.2** | **61.0** | **60.0** | **56.3** | 56.5 | **76.3** | **16.6** | **17.8** | **16.8** | **12.8** | **53.0** |
| *Vision-Language Only* | | | | | | | | | | | | | |
| Qwen3-VL-Instruct | BASE | - | - | - | - | - | **50.0** | - | - | - | - | - | 53.0 |
| | + CITATION | 39.0 | 52.0 | 52.2 | 52.0 | 25.5 | 48.0 | 30.2 | 39.8 | 40.4 | 40.1 | 17.5 | **55.0** |
| | + POST-HOC ATTRIBUTION | **98.9** | **69.7** | **70.8** | **70.2** | **69.4** | **50.0** | **93.4** | **44.5** | **44.8** | **44.6** | **42.3** | 53.0 |
| Qwen3-VL-Thinking | BASE | - | - | - | - | - | 47.0 | - | - | - | - | - | 51.0 |
| | + CITATION | 38.5 | 56.2 | 56.8 | 56.1 | 30.8 | **49.0** | 23.2 | 14.8 | 16.4 | 15.1 | 7.6 | **60.0** |
| | + POST-HOC ATTRIBUTION | **76.6** | **58.3** | **59.5** | **58.9** | **48.2** | 47.0 | **54.3** | **31.2** | **31.9** | **31.5** | **18.9** | 51.0 |
| Molmo2 | BASE | - | - | - | - | - | 41.0 | - | - | - | - | - | 50.5 |
| | + CITATION | 69.1 | **49.0** | 55.3 | **50.2** | **39.7** | 40.0 | **82.6** | **20.9** | **24.5** | **21.4** | **19.3** | 44.3 |
| | + POST-HOC ATTRIBUTION | **75.0** | 37.4 | 40.8 | 38.3 | 33.2 | **41.0** | 66.4 | 14.4 | 17.8 | 15.0 | 11.4 | **50.5** |

*Table 14.* Full results with Program-aided results on WorldSense with Gemini-3-Flash.

| Variant | Cov. | Attr. P | Attr. R | Attr. F1 | MURGAT-S | Acc |
|---|---|---|---|---|---|---|
| BASE + CITATION | 95.9 | 64.0 | 69.7 | 66.5 | 64.4 | 66.2 |
| BASE + POST-HOC ATTRIBUTION | 95.1 | 68.8 | 75.2 | 71.4 | 69.2 | **67.0** |
| LOGIC DECLARATIVE | 96.2 | 75.2 | 78.5 | 76.7 | 74.3 | 61.0 |
| LOGIC IMPERATIVE | 97.3 | **77.7** | 79.9 | **78.7** | **76.4** | 60.0 |
| NARRATIVE DECLARATIVE | 97.7 | 71.8 | 73.5 | 72.5 | 70.9 | 55.7 |
| NARRATIVE IMPERATIVE | **99.0** | 71.2 | **80.7** | 75.0 | 74.2 | 58.6 |

*Table 15.* Full Reasoning Results with different thinking levels.

| Model | Method | WorldSense | | | | | | Video-MMMU | | | | | |
|---|---|---|---|---|---|---|---|---|---|---|---|---|---|
| | | Cov. | Prec. | Rec. | Attr. | MURGAT-S | Acc | Cov. | Prec. | Rec. | Attr. | MURGAT-S | Acc |
| Gemini-3-Flash | Minimal | **98.9** | **68.8** | **72.7** | **70.5** | **69.7** | 70.0 | **93.4** | 55.9 | 64.4 | 59.5 | 55.3 | 86.3 |
| | Low | 98.8 | 64.1 | 68.4 | 65.9 | 65.2 | **71.0** | 89.5 | 59.5 | 69.7 | 63.8 | **57.1** | 85.4 |
| | Medium | 96.3 | 62.9 | 68.5 | 65.4 | 63.8 | 65.0 | 86.3 | 58.4 | 70.6 | 63.6 | 55.6 | **91.5** |
| | High | 95.9 | 64.0 | 69.7 | 66.5 | 64.4 | 66.2 | 88.2 | **59.9** | 71.0 | **64.5** | 56.9 | 86.0 |
| Gemini-3-Pro | Low | 69.9 | 63.1 | 65.6 | 64.2 | 45.6 | 62.6 | 50.0 | 63.9 | 69.6 | 66.3 | 32.4 | 83.2 |
| | High | **78.3** | **63.6** | **66.6** | **64.9** | **51.7** | 70.0 | **63.4** | **64.6** | **71.3** | **67.3** | **41.8** | **86.0** |

*Table 16.* Correlation of metrics with human judgments. We report Pearson ($r$), Spearman ($\rho$), and Kendall ($\tau$) coefficients across Coverage, Attribution Precision, Attribution Recall, and MURGAT-SCORE. **Our** is obtained by our evaluation protocol. Best results are **bolded**.

| Metric | Coverage | | | Attr. Precision | | | Attr. Recall | | | MURGAT-SCORE | | |
|---|---|---|---|---|---|---|---|---|---|---|---|---|
| | $r$ | $\rho$ | $\tau$ | $r$ | $\rho$ | $\tau$ | $r$ | $\rho$ | $\tau$ | $r$ | $\rho$ | $\tau$ |
| Holistic | 0.38 | 0.33 | 0.27 | 0.39 | 0.39 | 0.31 | 0.43 | 0.41 | 0.33 | 0.35 | 0.39 | 0.31 |
| Disentangled | 0.58 | 0.54 | 0.45 | 0.32 | 0.33 | 0.26 | 0.49 | 0.50 | 0.40 | 0.45 | 0.52 | 0.40 |
| Disentangled (sent-level) | 0.76 | 0.75 | 0.62 | 0.54 | 0.56 | 0.42 | 0.50 | 0.51 | 0.38 | 0.58 | 0.59 | 0.45 |
| **Our** | **0.97** | **0.97** | **0.89** | **0.65** | **0.64** | **0.49** | **0.59** | **0.59** | **0.44** | **0.86** | **0.84** | **0.69** |

# D. Additional Results.

## D.1. Full Results

Table 13 presents the complete main results, while detailed performance metrics for reasoning and program-aided tasks are provided in Table 14 and Table 15, respectively. Additionally, full correlation metrics are documented in Table 16. Finally, we show the breakdown of attribution precision by modality in Table 17.

## D.2. Qualitative Analysis

**Gemini-3-Pro vs Gemini-3-Flash.** In Table 1, we observe that Gemini-3-Pro performs worse than Gemini-3-Flash in attribution. We show the example in Figure 6. Qualitative analysis reveals that model-specific performance is often dictated by a fundamental trade-off between narrative synthesis and grounding precision. Specifically, we find that while larger models like Gemini-3-Pro attempt more intricate reasoning and spatial synthesis to provide a cohesive description, they are frequently susceptible to "spatial hallucinations" and temporal misalignment. These errors typically occur when the model attempts to build a global context across multiple cuts or infer details not explicitly visible in the cited frame. In contrast, Gemini-2.5-Flash often achieves higher Attribution scores by adopting a minimalist, shot-by-shot descriptive strategy. By prioritizing direct, verifiable observations over high-level narrative context, the smaller model avoids the "contextualization trap" where reasoning overrules precise visual evidence. This suggests that the drive for narrative fluency in larger models can occasionally compromise the faithfulness of their grounding citations.

**Post-hoc Attribution.** Our analysis of post-hoc attribution reveals a divergent impact across perceptual and deductive benchmarks, as illustrated in Figure 7. In perceptual tasks like WorldSense, post-hoc attribution serves as a critical

*Table 17.* Attribution Precision (%) split by modality (Visual vs. Audio) and Combined. Numbers in parentheses indicate the total count of citations checked for that modality. BASE is excluded as it generates no citations.

| Model | Method | WorldSense | | | Video-MMMU | | |
|---|---|---|---|---|---|---|---|
| | | Visual | Audio | All | Visual | Audio | All |
| Gemini-2.5-Flash | + CITATION | **70.8** (3019) | 52.0 (1760) | **64.2** | **77.7** (1767) | **40.5** (1451) | **59.6** |
| | + POST-HOC | 62.1 (2833) | **56.5** (1989) | 60.9 | 53.1 (3163) | 33.5 (1864) | 42.5 |
| Gemini-3-Flash | + CITATION | 65.6 (3622) | 58.4 (2201) | 64.0 | **71.4** (1818) | **45.7** (1511) | **59.9** |
| | + POST-HOC | **68.3** (1411) | **63.9** (892) | **68.8** | 53.5 (2266) | 36.4 (1796) | 43.6 |
| Gemini-3-Pro | + CITATION | **66.4** (1314) | 58.2 (809) | 63.6 | **72.8** (1200) | 41.5 (563) | **64.6** |
| | + POST-HOC | 65.5 (2106) | **63.4** (1491) | 65.4 | 57.8 (2443) | **42.9** (1636) | 41.0 |
| Qwen3-Omni-Instruct | + CITATION | **65.1** (545) | **53.1** (246) | **53.2** | **30.5** (945) | **12.6** (313) | **22.0** |
| | + POST-HOC | 45.6 (3422) | 39.0 (513) | 45.7 | 19.8 (8605) | 9.8 (1968) | 17.9 |
| Qwen3-Omni-Thinking | + CITATION | **65.3** (1481) | **51.2** (1333) | 56.4 | 14.0 (471) | 6.3 (337) | 7.8 |
| | + POST-HOC | 62.6 (1454) | 50.2 (963) | **59.2** | **20.6** (1675) | **7.5** (806) | **16.6** |
| *Vision-Language Only* | | | | | | | |
| Qwen3-VL-Instruct | + CITATION | 68.1 (516) | **58.5** (58) | 52.0 | **55.9** (551) | 1.8 (18) | 39.8 |
| | + POST-HOC | **70.0** (1461) | 47.2 (91) | **69.7** | 44.6 (2596) | **25.0** (11) | **44.5** |
| Qwen3-VL-Thinking | + CITATION | **77.0** (512) | 51.9 (72) | 56.2 | **36.8** (401) | **20.6** (185) | 14.8 |
| | + POST-HOC | 61.3 (1111) | **53.6** (87) | **58.3** | 35.6 (3303) | 14.7 (572) | **31.2** |
| Molmo2 | + CITATION | **57.4** (2589) | **45.1** (406) | **49.0** | **25.9** (3371) | **14.5** (259) | **20.9** |
| | + POST-HOC | 41.6 (2968) | 42.8 (333) | 37.4 | 20.0 (2475) | 6.6 (1406) | 14.4 |

mechanism for multimodal reinforcement. Decoupling the initial generation from the grounding process allows the model to perform a second perceptual pass that captures granular scene elements overlooked during the initial reasoning. This improves attribution recall and ensures the descriptive narrative is fully grounded. Conversely, on knowledge-intensive benchmarks like VideoMMMU, the post-hoc process introduces grounding overhead that compromises precision. Because the model relies on internal domain knowledge to solve complex problems, the subsequent attribution step forces a mapping of logical deductions to the visual stream. This results in performative citation, where the model anchors technical facts to generic introductory frames or irrelevant diagrams. These results indicate that while post-hoc attribution effectively grounds omnimodal perception, it introduces faithfulness noise in deductive tasks by incentivizing the model to fabricate visual evidence for internal reasoning steps.

**Program-Aided Generation.** We present a comparison of program-aided variants in Figure 8, where performance is largely governed by the interaction between execution style and synthesis logic. As presented in Table 14, an accuracy-attribution gap is observed in the Logic Imperative variant; despite achieving the highest attribution (78.7 F1), its accuracy (60.0) remains lower than the BASE + POST-HOC variant (67.0). This suggests that program-aided models can become "distracted" by the verification process—finding correct evidence but failing to synthesize it accurately during the final step—whereas the base models benefit from a holistic view without the noise of intermediate outputs. In contrast, Narrative Imperative excels in Recall/Coverage (80.7/99.0). Its instructional nature forces the model to execute specific actions, while the narrative style removes strict logical constraints, resulting in a "chatty" output that observes nearly all scene elements but lacks the precision to filter irrelevant noise. Finally, LOGIC DECLARATIVE offers the most stable performance across program-aided variants, with high precision (75.2) and balanced accuracy (61.0). By defining specific facts to be checked rather than open-ended instructions, Declarative prompts minimize the "trace drifting" common in long Imperative executions, ensuring that grounding remains focused and faithful to the task. We note it is difficult to balance attribution with accuracy.

# E. Prompts

## E.1. Automatic Evaluation

We provide the prompts used for atomic fact decomposition in Figure 9 and decontextualization in Figure 10. The prompt for verifiability evaluation can be found in Figure 12, Figure 13, and Figure 14 for Simple, CoT, and JSON variant, respectively. Similarly, the prompts for attribution entailment is in Figure 15, Figure 16, and Figure 17.

**Example 1**

| **Gemini-2.5-Flash** (Score: **1.0**) | **Gemini-3-Pro** (Score: **0.61**) |
|---|---|
| "...boy with dreadlocks... introduces the song by saying, *'This is called, song to you'* (**audio, 0:06-0:07**)." | "...male character... states, *'This is called 'Song To You''* (**audio, 0:06**)." |

| Model | Atomic Fact (Claim) | Cite | Judg. | Failure Mode |
|---|---|---|---|---|
| **Flash** | "This is called, song to you" | 0:06-0:07 | ✓ | Perfect timing. |
| **Pro** | "This is called 'Song To You'" | 0:06 | χ | **Temporal Miss:** Utterance lasts 1.5s; 0:06 is just the start. |

**Example 2**

| **Gemini-2.5-Flash** (Score: **0.47**) | **Gemini-3-Pro** (Score: **0.09**) |
|---|---|
| "A man wearing a white t-shirt is shown speaking into a microphone (**visual, 0:06**)." | "The video depicts two men sitting at a table equipped with microphones... (**visual, 0:06**)." |

| Model | Atomic Fact (Claim) | Cite | Judg. | Failure Mode |
|---|---|---|---|---|
| **Flash** | A man in white t-shirt is shown. | 0:06 | ✓ | Correct single-shot description. |
| **Pro** | "Two men are sitting at a table" | 0:06 | χ | **Spatial Hallucination:** Only one person visible in frame. |

*Figure 6.* Comparative analysis of Gemini 2.5 Flash and Gemini 3 Pro. While Pro attempts higher-level narrative synthesis (e.g., spatial layouts and song titles), it suffers from lower grounding precision compared to Flash's minimalist, observation-first approach.

## E.2. Response Generation

We provide the prompt used for generating the baseline output in Figure 18, the prompt for generating with citation in Figure 11, and the prompt for running post-hoc refinement in Figure 19.

**WorldSense: Post-hoc Attribution Fixes Missing Recall**

| **BASE + CITATION** (Recall Failure) | **Post-hoc Attribution** (Grounded) |
|---|---|
| "A woman wearing *blue overalls* prepares the soil in a *wooden planter*. She then plants the seeds at a depth of two inches **(visual, 0:22)**." | "A woman wearing *blue overalls* **(visual, 0:03)** prepares the soil in a *wooden planter* **(visual, 0:08)**. She then plants the seeds **(visual, 0:22)**..." |

| Method | Atomic Fact | Cite | Judg. | Outcome |
|---|---|---|---|---|
| **BASE + CITATION** | "Woman wearing blue overalls" | None | χ | **Low Recall:** Missed character grounding. |
| **Post-hoc** | "Woman wearing blue overalls" | 0:03 | ✓ | **Improved Recall:** Anchors initial scene elements. |

**VideoMMMU: Post-hoc Over-citation Leading to Precision Loss**

| **BASE + CITATION** (Precise) | **Post-hoc Attribution** (Hallucinated Mapping) |
|---|---|
| "The circuit reaches steady state; the *current through the inductor is 2A* as shown on the oscilloscope **(visual, 3:45)**." | "The *circuit* **(visual, 0:10)** reaches *steady state* **(audio, 1:05)**; the *current* **(visual, 1:20)**... is 2A **(visual, 3:45)**." |

| Method | Atomic Fact | Cite | Judg. | Failure Mode |
|---|---|---|---|---|
| **BASE + CITATION** | "Current... is 2A" | 3:45 | ✓ | Correct attribution to the measurement. |
| **Post-hoc** | "The current [is present]" | 1:20 | χ | **Context Mismatch:** 1:20 shows a *diagram* of a battery, not the live current measurement. |

*Figure 7.* Comparison of attribution strategies. On WorldSense, Post-hoc Attribution improves Recall by grounding descriptive scene elements missed by the Base model. Conversely, on VideoMMMU, the Post-hoc pass often results in "Citation Salad," incorrectly mapping specific technical steps to generic introductory frames.

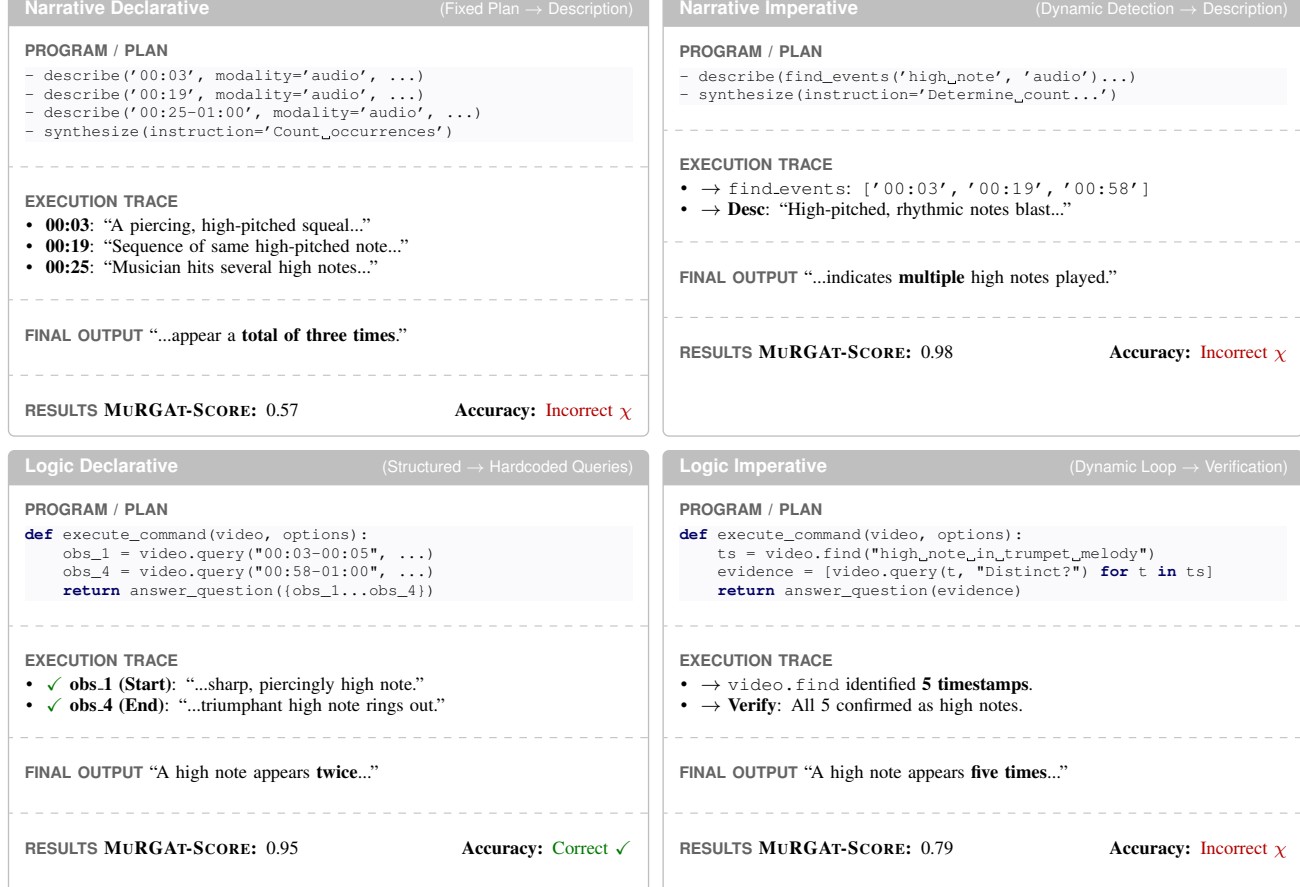

*Figure 8.* Qualitative comparison of four program-aided generation variants. **Narrative** variants struggle with exact quantification due to hallucinated or vague counts. **Logic Declarative** succeeds by sampling known logical intervals. **Logic Imperative** fails due to error propagation (over-counting candidates).

**Prompt for Atomic Decomposition**

**Role:** You are an expert Annotator for multimodal datasets.
**Task:** Break down the provided **Source Sentence** into a list of independent, self-contained atomic facts.
**Definitions:**

1. **Atomic Fact:** A short, standalone sentence containing a singular piece of information (e.g., an action, an object's presence, a specific property, or a quantity).

2. **Citations:** Parenthetical references like (visual, 0:00) or (audio, 1:05).

**Critical Rules:**
1. **Meta-Talk & Metadata Removal:**

   - **Remove** navigational phrases that describe the video medium rather than the content.

   - *Remove:* "The video shows," "The audio contains," "We can see," "The narrator states," "In the first example."

   - *Keep:* The actual content shown or stated.

   - **Example:** "The video shows a boy holding a guitar (visual, 0:05)." → "- A boy is holding a guitar (visual, 0:05)."

   - **Example:** "The narrator says 'Hello' (audio, 0:10)." → "- A person says 'Hello' (audio, 0:10)."

   - **Ignore Metadata:** Do not create facts about the video structure itself (e.g., ignore "The clip ends at 0:55" unless it's relevant to the narrative content).

2. **Adherence to Original Text:**

   - Adhere strictly to the original wording for technical terms, formulas, values, and equations. Do not reformat or interpret them (e.g., keep LaTeX or math symbols exactly as they appear in the source).

3. **Granularity (Split Adjectives & Actions):**

   - Split compound properties. "The music is lyrical and flowing" becomes two facts: one for "lyrical", one for "flowing".

   - Split compound actions. "He runs and jumps" becomes two facts.

4. **Citation Logic:**

   - **Propagation:** If the source sentence has a citation, **every** resulting atomic fact must inherit it.

   - **Splitting:** If citations are embedded (e.g., "A (visual, 1:00) hits B (visual, 2:00)"), assign the specific citation only to the relevant fact.

   - **No Valid Citation:** If the source text contains no citations, do not add any. Output the facts without citations.

**Examples:**
*Input:* A male character with long dreadlocks, dressed in a pink button-down shirt and a black vest, stands at a microphone (visual, 0:06).
*Output:*

   - A male character is present (visual, 0:06).

   - The character has long dreadlocks (visual, 0:06).

   - The character is dressed in a pink button-down shirt (visual, 0:06).

   - The character is dressed in a black vest (visual, 0:06).

   - The character stands at a microphone (visual, 0:06).

*Input:* The video states that product costs include direct material, direct labor, and overhead (visual, 0:15-0:18; audio, 0:15-0:18).
*Output:*

   - Product costs include direct material (visual, 0:15-0:18; audio, 0:15-0:18).

   - Product costs include direct labor (visual, 0:15-0:18; audio, 0:15-0:18).

   - Product costs include overhead (visual, 0:15-0:18; audio, 0:15-0:18).

**Current Sentence:**
{sent}
**Output:**

*Figure 9.* Prompt for Atomic Decomposition.

**Prompt for Decontextualization**

**Role:** You are an expert Linguistic Editor specializing in video caption refinement.
**Task:** Rewrite the text below to resolve vague references (pronouns, generic nouns) with specific entity names, strictly adhering to chronological availability of information.
**Primary Directive: The ``Forward-Only'' Rule**
You must resolve references based **ONLY** on information established in the text *prior* to the sentence you are editing.

- **Forbidden:** Do not ``back-fill'' details. If Sentence 1 says ``A man enters'' and Sentence 2 says ``The doctor sits,'' you cannot change Sentence 1 to ``The doctor enters.'' (We didn't know he was a doctor yet).

- **Allowed:** If Sentence 1 introduces ``Jeff'' and Sentence 2 says ``He,'' you must change ``He'' to ``Jeff.''

**Strict Constraints:**

1. **Preserve Citations:** Keep every citation (e.g., (visual, 0:05), [audio, 0:03-0:08]) exactly where it appears in the text. Do not move or merge them.

2. **Verify Claims:** Do not add descriptive adjectives (like ``red car'', ``angry man'') unless that specific sentence or a *prior* one explicitly establishes that attribute.

3. **Minimalism:** Replace the pronoun with the closest sufficient noun (e.g., replace ``it'' with ``the creature'', not ``the giant one-armed red creature'' unless necessary for disambiguation).

**Input Text:**
{INPUT_TEXT}
**Output (Rewritten Text):**

*Figure 10.* Prompt for Decontextualization.

---

**Prompt for Baseline Generation**

```
Carefully watch the provided video and listen strictly to the corresponding audio.
Your task is to select the best option that answers the question, based exclusively
on the provided content.
Before stating your final answer, you must provide a step-by-step reasoning process.
Strict Citation Rules:

 1. Mandatory Citations: Every single sentence containing a factual claim or
    observation must end with a specific citation in parentheses.

 2. Narrative vs. Timestamp:

        • Do NOT include specific numeric timestamps (e.g., ``at 0:15'') inside the
          narrative text.
        • DO describe the events using relative temporal language if needed (e.g., ``At
          the beginning'').
        • The numeric timestamp belongs only inside the parenthetical citation.

 3. Citation Format: Use (modality, timestamp).

        • Modality: visual or audio.
        • Timestamp: MM:SS (specific) or MM:SS-MM:SS (ranges).

 4. Combined Evidence: If multiple pieces of evidence are needed, separate them with a
    semicolon inside the same parentheses.

Examples of Correct vs. Incorrect Formatting:

  • Incorrect: ``From 0:50 onwards, the melody continues...''

  • CORRECT: ``Towards the end, the melody continues with sustained notes (audio,
    0:50-0:55).''

  • CORRECT (Multiple): ``The man points while speaking (visual, 0:12; audio,
    0:12-0:14).''

Output Format:
Reasoning: [Your step-by-step reasoning following the strict citation rules above]
Answer: [Only the letter of the correct option]
Question: {question}
{options}
```

*Figure 11.* Prompt for Baseline Generation with Citation.

---

**Prompt for Verification Worthiness (Simple)**

```
You are an expert evaluator for Multimodal Grounding. Your task is to determine if the
Sentence contains CHECK-WORTHY information.
INPUTS:

 1. Sentence: The text generation to evaluate.

GUIDELINES:
Output YES (Check-Worthy) if the sentence describes ANY specific, verifiable content in
the video/audio (actions, objects, text, specific values).
Output NO (Not Check-Worthy) if the sentence consists ENTIRELY of:

 1. Metadata/Reasoning: References to options (A, B, C), logical conclusions (starts
    with ``Therefore'', ``Thus''), or conditional logic without new visual claims.

 2. General Knowledge: Definitions or universal truths (e.g., ``Paris is in France'').

 3. Subjective: Opinions, fillers, or navigational text.

TASK:
Sentence: {sentence}
OUTPUT:
Output only the word YES or NO.
```

*Figure 12.* Prompt for Verification Worthiness (Simple Binary).

---

**Prompt for Verification Worthiness (CoT)**

```
You are an expert evaluator for Multimodal Grounding.  Your task is to determine if the
Sentence contains CHECK-WORTHY information.
DEFINITIONS:

  • CHECK-WORTHY (YES): The sentence contains specific visual/audio events, specific
    text on screen, or specific negative claims (what is missing).

  • NOT CHECK-WORTHY (NO): The sentence consists ENTIRELY of:

    1. Reasoning/Metadata:  Logical connectors (e.g., ``Therefore'', ``Thus''),
       references to ``Options'' or ``Statements'', or conditional logic.
    2. General Knowledge:  Universal truths not specific to this video.
    3. Subjective:  Opinions or conversational fillers.

TASK:
Sentence:  {sentence}
INSTRUCTIONS:

 1. Analyze the Sentence.  Does it describe any specific visual or audio details?

 2. If it contains any verifiable claim (even mixed with reasoning), mark it as YES.

 3. Only mark it as NO if it is purely structural, logical, or opinion-based without
    new visual information.

OUTPUT FORMAT:
Reasoning:  [Analyze the sentence content.]
Answer:  [YES or NO]
```

*Figure 13.* Prompt for Verification Worthiness (Chain-of-Thought).

---

**Prompt for Verification Worthiness (JSON)**

```
You are an expert evaluator for Multimodal Grounding.  Classify if the Sentence
contains CHECK-WORTHY information.
GUIDELINES:

  • YES: The sentence describes ANY specific, verifiable content in the video/audio
    (actions, objects, quantities, text, visual attributes).

  • NO: The sentence consists ENTIRELY of metadata (e.g., ``Option A is correct''),
    reasoning (e.g., ``Therefore, it matches''), general knowledge, or subjective
    opinions.

TASK:
Sentence:  {sentence}
OUTPUT FORMAT:
Return a single JSON object:
{
"reasoning":  "string (Explain if the sentence contains visual claims...)",
"label":  "string (YES or NO)"
}
```

*Figure 14.* Prompt for Verification Worthiness (JSON Output).

---

**Prompt for Atomic Entailment (Simple)**

```
You are an expert evaluator for Multimodal Grounding.  Determine if the provided Media
Content entails the Atomic Fact.
GUIDELINES:

  • YES (Supported):  The provided media segments (images/audio) contain clear evidence
    that fully supports the fact.

  • NO (Not Supported):  The media contradicts the fact, or the necessary information
    is missing from the provided segments.

TASK:
Media Content:  {context}
Atomic Fact:  {fact}
OUTPUT:
Output only the word YES or NO.
```

*Figure 15.* Prompt for Entailment (Simple Binary).

---

**Prompt for Atomic Entailment (CoT)**

You are an expert evaluator for Multimodal Grounding.  Determine if the provided **Media Content** entails the **Atomic Fact**.
**INPUTS:**

  • **Media Content:**  A set of video frames, audio segments, or images.

  • **Atomic Fact:**  The statement to verify.

**INSTRUCTIONS:**

 1. **Observation:**  Examine ALL provided media segments.  Describe what is visible or audible relevant to the fact.

 2. **Verification:**  Compare your observations to the specific details in the Atomic Fact (actions, colors, values, timing).

 3. **Judgment:**

      • Return **YES** only if the evidence is present and precise.
      • Return **NO** if the evidence is missing, ambiguous, or contradictory.

**TASK:**
Atomic Fact:  {fact}
**OUTPUT FORMAT:**
Reasoning:  [Describe evidence found in the media and compare it to the fact.]
Answer:  [YES or NO]

*Figure 16.* Prompt for Entailment (Chain-of-Thought).

---

**Prompt for Atomic Entailment (JSON)**

You are an expert evaluator for Multimodal Grounding.  Verify if the **Atomic Fact** is supported by the **Media Content**.
**GUIDELINES:**

  • **YES:** Strong evidence exists in the media.

  • **NO:** Evidence is missing, unrelated, or contradictory.

**TASK:**
Atomic Fact:  {fact}
**OUTPUT FORMAT:**
Return a single JSON object:
{
"evidence_description":  "string (Briefly describe what is seen/heard...)",
"label":  "string (YES or NO)"
}

*Figure 17.* Prompt for Entailment (JSON Output).

---

**Prompt for Baseline Generation**

Carefully watch the provided video and listen strictly to the corresponding audio.
Your task is to select the best option that answers the question, based **exclusively** on the provided content.
Before stating your final answer, you must provide a step-by-step reasoning process.
**Output Format:**
Reasoning:  [Your step-by-step reasoning]
Answer:  [Only the letter of the correct option]
Question:  {question}
{options}

*Figure 18.* Prompt for Baseline Generation (No Citations).

**Prompt for Post-hoc Attribution Correction**

You are a rigorous Quality Assurance Editor for multimodal video analysis. Your task is to review a provided model output, critically analyze the citations for accuracy and formatting, and apply fixes where necessary.
**Input Context:**

 1. **Video/Audio Content**

 2. **Model Output to Review**: {{Output}}

**Your Task:**
Review the ''Model Output'' and produce a **Revised Output**. You must correct errors related to citation formatting, citation placement, and entailment (evidence accuracy).
**Strict Editing Rules (Do NOT deviate):**

 1. **Preserve Narrative Text**: Do **not** rewrite, summarize, or alter the reasoning text or the final answer choice. Your job is *only* to fix the mechanics of the citations and remove timestamps from the prose.

 2. **Fix Citation Format**: Ensure every citation follows the exact format: (modality, timestamp).

      • *Correct:* (visual, 0:15), (audio, 0:10-0:15), (visual, 0:12; audio, 0:14).
      • *Incorrect:* [0:15], (Video, 0:15), (0:15-0:20).

 3. **Fix Timestamp Placement:**

      • If a numeric timestamp (e.g., ''At 0:15...'') appears in the narrative text, **remove it** and ensure it is properly placed in the parenthetical citation at the end of the sentence.
      • Keep relative temporal words (e.g., ''At the start,'' ''Later'') in the text.

 4. **Verify Entailment & Hallucination:**

      • Check if the cited timestamp actually supports the claim made in the sentence.
      • If a citation is missing for a factual claim, add the correct (modality, timestamp) based on the video evidence.

**Output Structure:**
Return the full text with the corrections applied. Do not add conversational filler. Just provide the final cleaned Reasoning and Answer.

*Figure 19.* Prompt for Post-hoc Citation Attribution and Correction.

