# OpenReview forum: "Multimodal Fact-Level Attribution for Verifiable Reasoning"
_ICML.cc/2026/Conference — ICML 2026 regular_

### Official Review · Reviewer_DqLU · 2026-03-07

**Soundness:** 3
**Presentation:** 3
**Significance:** 2
**Originality:** 2
**Overall Recommendation:** 4
**Confidence:** 4

**Summary:**

MURGAT is a benchmark + automatic evaluation pipeline for fact-level multimodal attribution in MLLMs. Given video, audio, and figure inputs, models must generate answers with citations that specify both modality and exact timestamp. A three-stage pipeline (verifiable claim identification, atomic fact decomposition, attribution entailment) yields MURGAT-SCORE, which achieves r=0.84 correlation with human judgments. Key finding: models can answer correctly while hallucinating citations, and more thinking effort actively degrades attribution quality.

**Compliance With Llm Reviewing Policy:**

Affirmed.

**Final Justification:**

It's a reasonable paper with clear score upper board to me. I think it's acceptable but i wont be disappointed if it gets rejected because it can only be interesting to a specific group of readers at most. Therefore, I didnt increase my score.

**Key Questions For Authors:**

NO

**Strengths And Weaknesses:**

**Pros**

- The task is interesting and underexplored. Fact-level attribution with temporal grounding across heterogeneous modalities (video + audio + graphs simultaneously) is harder than prior text-only benchmarks.
- The three-stage evaluation pipeline is solid and the ablations properly justify each design choice: explicit decontextualization, sentence-level vs. response-level decomposition, etc. Competent benchmark engineering.
- The reasoning-degrades-attribution finding (Figure 3) is counterintuitive and empirically supported. Higher thinking effort hurts MURGAT-SCORE while barely helping accuracy. That is a genuine empirical signal, not an obvious result.

---

**Cons**

- The "first" claim is wrong? MAVIS[1] introduced a 157K-instance benchmark for multimodal source attribution with fact-level citations, covering long-form VQA with document-level multimodal evidence. The paper should discuss its relative to MAVIS, which is the most direct competitor and has 3 orders of magnitude more data.
[1] Song et al., MAVIS: A Benchmark for Multimodal Source Attribution in Long-form Visual Question Answering

Minor:
- Human annotation is probably too small to validate the metric reliably. The entire empirical basis for the r=0.84 claim is 80 model outputs (20 examples × 4 models), yielding 1,421 atomic facts. Pearson r on 80 samples has massive variance.

- The benchmark scale and dataset release are unclear. The main generation experiments (Table 5) use 100 examples per dataset sampled from WorldSense and Video-MMMU, both of which already exist. The paper never states how many total videos are in MURGAT, whether it will be publicly released, or whether MURGAT is a new dataset at all versus an evaluation protocol on existing data.

---

> ### Author Rebuttal · Authors · 2026-03-30
>
> We thank the reviewer for recognizing that the task is **"interesting and underexplored,"** that the **"three-stage evaluation pipeline is solid,"** and that the reasoning-degrades-attribution finding is **"counterintuitive and empirically supported"** and **"a genuine empirical signal."** We address each question below.
>
> **\> W1. MAVIS comparison missing.**
> We would like to clarify that we do cite MAVIS (Song et al., 2025\) in our related work (L112-113). The key distinctions are: (1) MAVIS focuses on image-based VQA with document-level evidence; MURGAT targets video \+ audio \+ figures with temporal grounding. (2) MAVIS evaluates short, observation-based answers; MURGAT targets complex multi-step reasoning with verifiable vs. non-verifiable claim distinction. (3) MURGAT requires temporal citations (e.g., audio 0:42–0:46), adding a localization challenge absent in MAVIS.
>
> We will soften the "first" claim to "first benchmark for fact-level attribution with temporal grounding across heterogeneous modalities" and add a detailed MAVIS comparison in Related Work.
>
> **\> W2. Human annotation sample too small (80 outputs \-\> r=0.84 has high variance)**
> We would like to clarify that while we sample 20 input examples, the actual annotation volume is substantially larger due to the multi-model, multi-stage design. Specifically, we evaluate 4 models on each input, yielding 80 model-generated responses. The human annotations scale at each stage of the pipeline: Stage 1 (Verifiable Claim Identification) annotates 600 sentences, producing 580 verifiability labels; Stage 2 (Atomic Fact Decomposition) annotates these sentences into 2,032 atomic facts across 635 decompositions; Stage 3 (Attribution Quality) evaluates 917 entailment labels from the resulting 1,421 verifiable atomic facts. These per-subtask validation sets are comparable to prior work (e.g., Jacovi et al. 2025, \\\~500 examples). The 100-example main experiments use the automated pipeline (validated against these human judgments) to evaluate thousands of atomic facts across multiple models.
>
> To address statistical reliability, we conducted bootstrap significance testing (10,000 resamples). **All correlations across all metrics and evaluator variants are significant at p \< 0.001**, with tight 95% CIs:
>
> | Evaluator (Subtask 1–2 / 3\) | Metric | Spearman | 95% CI |
> | :---- | :---- | :---- | :---- |
> | Ours | final\_score | 0.844 | \[0.737, 0.911\] |
> | Ours | coverage | 0.967 | \[0.939, 0.983\] |
> | GPT-5 / G2.5-Flash | final\_score | 0.776 | \[0.641, 0.868\] |
> | G2.5-Flash | final\_score | 0.836 | \[0.722, 0.910\] |
> | G3-Pro | final\_score | 0.853 | \[0.744, 0.921\] |
>
> Bootstrap subsampling further shows correlations are stable even with fewer examples — the median Spearman remains \~0.84 across all sample sizes, with CI width narrowing monotonically:
>
> | Sample Size | Median Spearman | 95% CI | CI Width |
> | :---- | :---- | :---- | :---- |
> | 30 | 0.841 | \[0.642, 0.940\] | 0.298 |
> | 50 | 0.842 | \[0.700, 0.924\] | 0.223 |
> | 80 | 0.843 | \[0.737, 0.911\] | 0.174 |
>
> **\> W3. Benchmark scale and dataset release unclear**
> We will release: (1) evaluation code and pipeline, (2) all human annotations (580 verifiability, 635 decomposition, 917 entailment labels), (3) model outputs for all evaluated models, and (4) evaluation splits (100 examples per dataset). We will clarify that MURGAT is an evaluation protocol applied to existing datasets, supplemented by curated human annotations.

---

> > ### Author Rebuttal · Reviewer_DqLU · 2026-04-01
> >
> > Thank you for the clarifications. I am maintaining my score, as the rebuttal consists primarily of clarifications rather than new evidence.

---

> > > ### Author Response · Authors · 2026-04-06
> > >
> > > We thank the reviewer for the **engagement and for their positive assessment of the task, pipeline design, and empirical findings**. We respectfully note that the rebuttal included several pieces of new evidence beyond clarifications:
> > > 1. Bootstrap significance testing (10,000 resamples) with 95% CIs showing all correlations significant at p < 0.001 and stable across sample sizes
> > > 2. Alternative evaluator configurations (GPT-5, Gemini-only variants) with correlation comparisons
> > > 3. 5 newly evaluated models across three families with full results on both datasets
> > > 4. A concrete release plan for code, annotations, and model outputs.
> > >
> > > We will incorporate these results and the expanded MAVIS comparison into the revised paper. We hope this allow you to revisit your score.

---

### Official Review · Reviewer_odX8 · 2026-03-11

**Soundness:** 2
**Presentation:** 2
**Significance:** 2
**Originality:** 2
**Overall Recommendation:** 3
**Confidence:** 3

**Summary:**

The authors propose MURGAT, a new benchmark designed to evaluate how well Multimodal Large Language Models (MLLMs) can ground their reasoning with precise, fact-level citations across video, audio, and other modalities. Their findings reveal that even top-performing models frequently hallucinate citations, exposing a critical trade-off where deeper reasoning often undermines verifiable attribution.

**Compliance With Llm Reviewing Policy:**

Affirmed.

**Final Justification:**

The rebuttal helps with clarifying some of the previous misunderstandings. However, I still feel the originality/novelty of the contribution is not substantial enough for ICML. So I'm raising the score from 2 to 3.

**Key Questions For Authors:**

In Section 5.1 the paper mentioned that “We evaluate on Video-MMMU and WorldSense” and “we evaluate five representative models capable of handling both modalities: Gemini-2.5-Flash, Gemini-3-Flash, Gemini-3-Pro, Qwen3-Omni-Instruct, and Qwen3-Omni-Thinking.” But I didn’t see the results from Qwen3-Omni-Instruct, and Qwen3-Omni-Thinking on Video-MMMU. Also I noticed the Table 5 and Table 9 are split. I think the author can combine them for convenient reading.

**Limitations:**

see weakness

**Strengths And Weaknesses:**

Strength:
1. An evaluation framework to judge the LLM response. This framework combined the audio and video information. The framework includes a score metric to better evaluate the LLM response coverage and accuracy.
2. Motivation is clear, verifying the reasoning process in LLM is important.

Weakness:
1. LLM-as-Judge, especially in visual tasks: it is hard for current VLM to verify the image and text without ground-truth. Even the most advanced visual models today can produce hallucination. But the core part of MURGAT comes from LLM-as-Judge. The author compared different models' accuracy in Judgment and tested different prompts. It is still not enough to solve the gap in judgement accuracy, especially without ground truth for LLM-as-Judge to refer.
2. The author mentioned that “To capture diverse model behaviors, we randomly sampled 10 examples from each of two datasets”. I think 20 samples are not enough to evaluate the model performance. The evaluation results may contain bias, hence do not accurately reflect the model's performance. In addition, in section 5.1 “We evaluate on Video-MMMU and WorldSense, sampling 100 examples distinct from the human annotation set.” for the main experiment part 100 examples may also have bias.
3. The paper presents results on Gemini-2.5-Flash, Gemini-3-Flash and Gemini-3-Pro's performance. But the LLM-as-Judge process is also based on these three models: Gemini-2.5-Flash, Gemini-3-Flash and Gemini-3-Pro in Table 5. This may introduce bias.

---

> ### Author Rebuttal · Authors · 2026-03-30
>
> We thank the reviewer for appreciating the new **"evaluation framework"** that **"combined the audio and video information"** and the **"clear"** motivation for **"verifying the reasoning process in LLM."** We address each question below.
>
> **\> W1. LLM-as-Judge reliability without ground truth**
> We would like to clarify that our LLM-as-Judge does not operate in a reference-free setting — for entailment verification, the judge is provided with the original source material (the exact video frames/audio segments at cited timestamps) as ground truth, and simply determines whether these sources entail the atomic fact. This is fundamentally a verification task against concrete evidence, analogous to NLI-based evaluation (Laban et al., 2022, L253), rather than open-ended judgment without an anchor. We validate the judge against human ground-truth annotations (r=0.84), following standard methodology in attribution evaluation (Jacovi et al., 2025; Liu et al., 2023a, L281-282). We additionally: (1) use human annotations as the authoritative evaluation for initial model comparison (Table 1), (2) select different models for different subtasks based on human correlation, and (3) provide all baselines so the community can swap judges as models improve.
>
> To further validate the reliability of our automated judge, we conducted a manual spot-check of 50 randomly sampled entailment judgments from the pipeline output. Of these, 47 (94%) were confirmed accurate by the authors upon reviewing the corresponding video/audio evidence alongside the atomic fact and citation. The 3 disagreements involved borderline cases where temporal boundaries were ambiguous. This high agreement rate provides additional confidence that the automated judge produces reliable assessments.
>
> **\> W2. Sample size too small (20 for human annotation, 100 for experiments)**
> We would like to clarify that while we sample 20 input examples, the actual annotation volume is substantially larger due to the multi-model, multi-stage design. Specifically, we evaluate 4 models on each input, yielding 80 model-generated responses. The human annotations scale at each stage of the pipeline: Stage 1 (Verifiable Claim Identification) annotates 600 sentences, producing 580 verifiability labels; Stage 2 (Atomic Fact Decomposition) annotates these sentences into 2,032 atomic facts across 635 decompositions; Stage 3 (Attribution Quality) evaluates 917 entailment labels from the resulting 1,421 verifiable atomic facts. These per-subtask validation sets are comparable to prior work (e.g., Jacovi et al. 2025, \\\~500 examples). The 100-example main experiments use the automated pipeline (validated against these human judgments) to evaluate thousands of atomic facts across multiple models.
>
> To address statistical reliability, we conducted bootstrap significance testing (10,000 resamples). **All correlations across all metrics and evaluator variants are significant at p \< 0.001**, with tight 95% CIs:
>
> | Evaluator (Subtask 1–2 / 3\) | Metric | Spearman | 95% CI |
> | :---- | :---- | :---- | :---- |
> | Ours | final\_score | 0.844 | \[0.737, 0.911\] |
> | Ours | coverage | 0.967 | \[0.939, 0.983\] |
> | GPT-5 / G2.5-Flash | final\_score | 0.776 | \[0.641, 0.868\] |
> | G2.5-Flash | final\_score | 0.836 | \[0.722, 0.910\] |
> | G3-Pro | final\_score | 0.853 | \[0.744, 0.921\] |
>
> Bootstrap subsampling further shows correlations are stable even with fewer examples — the median Spearman remains \~0.84 across all sample sizes, with CI width narrowing monotonically:
>
> | Sample Size | Median Spearman | 95% CI | CI Width |
> | :---- | :---- | :---- | :---- |
> | 30 | 0.841 | \[0.642, 0.940\] | 0.298 |
> | 50 | 0.842 | \[0.700, 0.924\] | 0.223 |
> | 80 | 0.843 | \[0.737, 0.911\] | 0.174 |
>
> **\> W3. Self-evaluation bias (Gemini evaluating Gemini)**
> Due to space constraints, please refer to our response to **Reviewer 3sXw W2** for the full analysis. In summary: (1) no single model evaluates its own outputs end-to-end; (2) we ran alternative evaluator configurations (including GPT-5) and all achieve strong correlations (p\<0.001); (3) Gemini evaluators do not preferentially inflate Gemini outputs — the G3-Pro evaluator scores its own outputs *lower* than G2.5-Flash.
>
> **\> Q1. Missing Qwen results on Video-MMMU; combine Tables 5 and 9**
> We will merge Tables 5 and 9\. Due to space constraints, please refer to our response to **Reviewer rsJJ W1** for the full expanded evaluation. We now include 5 newly evaluated models (Qwen-Omni-Instruct/Thinking, Qwen-3-VL-Instruct/Thinking, Molmo2) in addition to the existing Gemini results. Qwen-Omni models achieve substantially lower MURGAT-S than Gemini across both datasets. We observe all models find the task difficult, confirming the attribution challenges are not Gemini-specific.

---

> > ### Author Rebuttal · Reviewer_odX8 · 2026-04-01
> >
> > Thanks for the clarification. I'll raise my score to 3.

---

> > > ### Author Response · Authors · 2026-04-06
> > >
> > > We thank the reviewer for the **constructive engagement and for confirming that the concerns have been adequately addressed**. We would like to point out the key additions during the rebuttal were:
> > > 1. A manual spot-check of 50 entailment judgments (94% accuracy) addressing the LLM-as-Judge concern
> > > 2. Bootstrap significance testing with 95% CIs confirming statistical reliability
> > > 3. Alternative evaluator configurations including GPT-5 addressing self-evaluation bias
> > > 4. 5 newly evaluated models (Qwen-Omni, Qwen-VL, Molmo2) with full results on both datasets.
> > >
> > > We will incorporate all of these into the revised paper.

---

### Official Review · Reviewer_rsJJ · 2026-03-12

**Soundness:** 4
**Presentation:** 3
**Significance:** 3
**Originality:** 4
**Overall Recommendation:** 3
**Confidence:** 5

**Summary:**

Prior MLLM benchmarks are limited to a narrow set of modalities, which overlooks other modalities. To evaluate MLLM reasoning ability in heterogeneous multimodal inputs, the author proposed MUGAT to evaluate MLLM reasoning ability for fact-level multimodal attribution. Finding out though current SOTA MLLMs achieve high performance in question-answering, they fail in multimodal attribution.

**Compliance With Llm Reviewing Policy:**

Affirmed.

**Key Questions For Authors:**

Questions:
1. Limited model for evaluation. The author only uses the Gemini series model for evaluation. There are a lot of other omni-models, such as Qwen-omni and Phi. So there is a need to evaluate other models to support the finding.
2. For human annotation, there is a need to provide a detailed description for annotation.
3. There is a need for error analysis. It could be better to provide detailed analysis for Gemini's failed cases for future research.

**Limitations:**

Yes

**Strengths And Weaknesses:**

Strength:
1. First benchmark for evaluate fact-level attribution for multimodal large language models
2. Proposed MURGAT-Score for fine-grained automatic evaluation pipeline
3. Finding out SOTA MLLMs don't have capability to ground.

---

> ### Author Rebuttal · Authors · 2026-03-30
>
> We thank the reviewer for recognizing MURGAT as the **"first benchmark to evaluate fact-level attribution for multimodal large language models"** and the **"fine-grained automatic evaluation pipeline."** We address each question below.
>
> **\> W1. Limited model diversity — only Gemini evaluated**
>
> As the reviewer suggested, we have evaluated Qwen-Omni models (Instruct, Thinking). We further assessed how vision-language (VL) models that lack native audio processing perform, evaluating Qwen-3-VL (Instruct, Thinking) and Molmo2-8B. The tables below present results (Gemini results are in the original Table 5).
>
> *WorldSense:*
>
> | Model | Method | Coverage | Attribution | MURGAT-S | Acc |
> | :---- | :---- | :---- | :---- | :---- | :---- |
> | Qwen-Omni-Instruct | BASE | \- | \- | \- | 57.0 |
> |  | \+ CITATION | 47.6 | 53.3 | 29.0 | 54.0 |
> |  | \+ POST-HOC | 99.5 | 45.7 | 45.4 | 57.0 |
> | Qwen-Omni-Thinking | BASE | \- | \- | \- | 56.5 |
> |  | \+ CITATION | 52.7 | 56.3 | 31.3 | 61.0 |
> |  | \+ POST-HOC | 93.2 | 60.0 | 56.3 | 56.5 |
> | Qwen-3-VL-Instruct | BASE | \- | \- | \- | 50.0 |
> |  | \+ CITATION | 39.0 | 52.0 | 25.5 | 48.0 |
> |  | \+ POST-HOC | 98.9 | 70.2 | 69.4 | 50.0 |
> | Qwen-3-VL-Thinking | BASE | \- | \- | \- | 47.0 |
> |  | \+ CITATION | 38.5 | 56.1 | 30.8 | 49.0 |
> |  | \+ POST-HOC | 76.6 | 58.9 | 48.2 | 47.0 |
> | Molmo2 | BASE | \- | \- | \- | 41.0 |
> |  | \+ CITATION | 69.1 | 50.2 | 39.7 | 40.0 |
> |  | \+ POST-HOC | 75.0 | 38.3 | 33.2 | 41.0 |
>
> *Video-MMMU:*
>
> | Model | Method | Coverage | Attribution | MURGAT-S | Acc |
> | :---- | :---- | :---- | :---- | :---- | :---- |
> | Qwen-Omni-Instruct | BASE | \- | \- | \- | 45.0 |
> |  | \+ CITATION | 34.6 | 21.8 | 9.8 | 40.0 |
> |  | \+ POST-HOC | 95.1 | 17.9 | 17.6 | 45.0 |
> | Qwen-Omni-Thinking | BASE | \- | \- | \- | 53.0 |
> |  | \+ CITATION | 36.3 | 7.6 | **4.8** | 51.0 |
> |  | \+ POST-HOC | 76.3 | 16.8 | 12.8 | 53.0 |
> | Qwen-3-VL-Instruct | BASE | \- | \- | \- | 53.0 |
> |  | \+ CITATION | 30.2 | 40.1 | 17.5 | 55.0 |
> |  | \+ POST-HOC | 93.4 | 44.6 | 42.3 | 53.0 |
> | Qwen-3-VL-Thinking | BASE | \- | \- | \- | 51.0 |
> |  | \+ CITATION | 23.2 | 15.1 | 7.6 | 60.0 |
> |  | \+ POST-HOC | 54.3 | 31.5 | 18.9 | 51.0 |
> | Molmo2 | BASE | \- | \- | \- | 50.5 |
> |  | \+ CITATION | 82.6 | 21.4 | 19.3 | 44.3 |
> |  | \+ POST-HOC | 66.4 | 15.0 | 11.4 | 50.5 |
>
> **Gemini vs. Qwen-Omni.** Qwen-Omni models achieve substantially lower MURGAT-S than Gemini across both datasets. The best Gemini (3-Flash, \+POST-HOC) reaches 69.2 on WorldSense vs. 56.3 for the best Qwen-Omni (Thinking, \+POST-HOC) — a 12.9-point gap. On Video-MMMU the gap widens: 56.9 vs. 17.6. This confirms attribution challenges are not Gemini-specific but amplified in other families.
>
> **Additional Key findings:**
>
> 1. Qwen-Omni-Thinking shows a unique failure: citations boost accuracy (+9.0% on Video-MMMU) but the model struggles with valid timestamp formats, yielding extremely low MURGAT-S (4.8 with \+CITATION).
> 2. Omni models outperform VL counterparts on audio-dependent WorldSense (57.0% vs 50.0%), but this reverses on Video-MMMU (45.0% vs 53.0%).
> 3. VL models hallucinate audio citations (up to 31.6% of references) despite lacking an audio encoder.
>
> **\> W2. Need detailed human annotation description**
> Detailed annotation procedures are in Appendix A: data (A.1), atomic decomposition (A.2), verifiable claim identification and attribution verification (A.3). We will add annotation UI screenshots in the camera-ready version.
>
> **\> W3. Need error analysis for Gemini's failed cases**
> We conducted a detailed qualitative analysis:
>
> **1\. Gemini-3-Pro vs. Gemini-2.5-Flash:** Pro attempts narrative synthesis but suffers from *spatial hallucinations* and *temporal misalignment*. For example, Pro cites (audio, 0:06) for a 1.5s utterance — catching only the start (X) — while Flash correctly cites (audio, 0:06-0:07) (V). Pro also hallucinates "two men" at a table when only one is visible (X), while Flash correctly describes a single person (V). **Takeaway:** Larger models' drive for fluency can compromise grounding faithfulness.
>
> **2\. Post-hoc attribution:** On recognition tasks (WorldSense), post-hoc fixes missing citations (V). On reasoning tasks (Video-MMMU), it creates "citation salad" by force-aligning abstract steps to generic frames (X). **Takeaway:** Post-hoc improves recall for perceptual tasks but hurts precision on reasoning tasks.
>
> We will include these analyses in detail in the final version.

---

> > ### Author Rebuttal · Reviewer_rsJJ · 2026-04-04
> >
> > The rebuttal addressed part of my concerns, but my main concern is still not fully resolved. Therefore, I maintain my score.

---

> > > ### Author Response · Authors · 2026-04-06
> > >
> > > We thank the reviewer for the engagement. All three weaknesses raised in the original review have been **substantively addressed with new models, data, and analysis**. We summarize below:
> > >
> > > **W1 (Model diversity):** Updated evaluation now includes 8 models across three families (Gemini, Qwen-Omni, VL-only baselines), with full results on both datasets provided in the rebuttal. New findings include Qwen-Omni-Thinking’s citation format failure mode and VL models hallucinating audio citations.
> > >
> > > **W2 (Annotation details):** Full protocol is in Appendix A of the submitted paper (L660–714), including guidelines, UI screenshots, inter-annotator agreement (73.7% verifiability, 86.1% attribution), timing statistics, and over-citation analysis.
> > >
> > > **W3 (Error analysis):** Qualitative analysis provided in the rebuttal covering Pro vs. Flash failure modes, post-hoc attribution divergence across task types, and a four-variant program-aided comparison.
> > >
> > > Since the reviewer indicates a concern remains but does not specify which, we hope this summary helps clarify. We are happy to provide additional information the reviewer or area chair may find helpful.

---

### Official Review · Reviewer_3sXw · 2026-03-12

**Soundness:** 2
**Presentation:** 2
**Significance:** 3
**Originality:** 3
**Overall Recommendation:** 4
**Confidence:** 2

**Summary:**

This paper introduces MURGAT (Multimodal Reasoning with Grounded Attribution), a benchmark designed to evaluate how well multimodal large language models (MLLMs) attribute their reasoning to specific evidence across diverse sources like video, audio, and graphs. It proposes a three-step evaluation protocol: verifiable claim identification, atomic fact decomposition, and attribution quality assessment, and introduces MURGAT-SCORE to quantify the fact-level attribution performance.

**Compliance With Llm Reviewing Policy:**

Affirmed.

**Final Justification:**

Although the authors have provided statistics on sentences with multiple sub-facts and sub-citations to argue that their proposed metric is unaffected on the test dataset, I still think that the design of the metric itself is inherently flawed. Nevertheless, in light of the authors' effort, I have raised my score from 3 to 4.

**Key Questions For Authors:**

- What is the specific prompting method used for Gemini in Figure 3? Is it the "+ CITATION" approach described in Table 5?

- Rather than claiming the proposed a new benchmark, it would be more accurate to say the authors propose a new evaluation framework/metric. Unlike conventional benchmarks, the paper does not introduce an new dataset. Instead, it establishes new evaluation criteria built upon existing datasets.

- I suggest the authors further refine the Introduction. The current version confuses readers to believe the motivation is to collect more specialized questions truly requiring multi-modalities to answer, to address the issue that existing benchmarks rely too much on internal model knowledge or single modalities. As the paper progresses, it becomes clear that the primary contribution is actually a new evaluation system rather than a new data collection.

If the author addresses my concerns about weaknesses and questions, I would raise my rating.

**Limitations:**

Yes.

**Strengths And Weaknesses:**

## Strengths
1. Novel Evaluation Perspective. The paper introduces MURGAT, a benchmark specifically designed for "fact-level attribution" in complex multimodal reasoning (video, audio, graphs), moving beyond simple observation-based tasks.

2. Rigorous Experimental Design. The study first uses human audits to prove citation hallucinations in mainstream MLLMs, then develops MURGAT-SCORE and an automated evaluation pipeline with high human correlation. Finally, it uses the automated pipeline to enable reliable, large-scale performance testing.

3. Insightful Observation of Model Flaws. It reveals a critical "reasoning tax" where forcing models to provide citations or increasing thinking length can actually decrease accuracy, highlighting a disconnect between reasoning and verifiable grounding.

## Weaknesses
1. Flaws in the Evaluation Metric. For instance, in Figure 1, the first sentence in the model response contains two atomic facts and two citations. It is penalized because one fact (F2) cannot be verified by one of the citations (visual). In practice, natural language sentences often combine multiple facts supported by different sources, so I think such structures should be acceptable rather than penalized.
﻿
2. Limitations of the Automated Evaluation System. The automated process shows relatively weak correlation with human judgment in precision and recall (~ 0.6), inspite of its high correlation in coverage (0.97), making the reliability of its precision/recall scores questionable. Furthermore, using the same models (like Gemini) to evaluate themselves without ground truth references is problematic, especially since the authors' own data in Table 1 proves these models have poor fact grounding capabilities.

---

> ### Author Rebuttal · Authors · 2026-03-30
>
> We thank the reviewer for recognizing MURGAT's **”novel evaluation perspective”**, the **”rigorous experimental design,”** and the **”insightful observation”** of the reasoning tax. We address each question below.
>
> **\> W1. Penalizing multi-source sentences**
> This behavior is by design: some examples require both modalities to answer, so the model must cite both audio and visual sources. We thus evaluate whether each citation supports the specific fact it is attached to. In the Figure 1 example, the sentence “The video explicitly defines the convention that 'repulsive forces are positive' on the graph (audio, 0:42-0:46; visual, 0:45)” carries two citations (**S1** `audio`, **S2** `visual`) and decomposes into **F1** *Repulsive forces are positive* and **F2** *A convention is defined*. Each fact is checked independently: F1 is supported (S1+S2 entails it); F2 is supported (S1 alone suffices). For precision, S2 is flagged as unnecessary for F2 since the visual frame alone does not entail “a convention being defined” (an auditory claim). Crucially, F1 receives full credit. The result is recall \= 1.0 (2/2) and precision \= 0.5 (1/2) — not a blanket zero. Only the specific fact-citation mismatch is penalized, consistent with text attribution benchmarks (Gao et al., 2023b; Liu et al., 2023a, L085-088). We will clarify this in Section 3.3.
>
> **\> W2. Weak precision/recall correlation (\~0.6); self-evaluation bias**
> **On correlation values:** The moderate precision (0.65) and recall (0.59) correlations in Table 4 are **not** due to poor entailment judgment — they arise from cascading differences in atomic fact decomposition. When the model generates a different *number* of atomic facts than the human annotator, the precision/recall denominators shift even if entailment predictions are perfect.
>
> Table 2 isolates entailment quality on *human-decomposed* facts, where precision, recall, and F1 are all **\~73** — comparable to Jacovi et al. (2025), who report 71.47 macro F1, and close to human inter-annotator agreement (86.1%, Appendix A.3). Holistically, MURGAT-SCORE achieves **r=0.86, Spearman=0.84, tau=0.69** (Table 12), reliably ranking model outputs at the response level, which is the granularity at which model comparison is performed.
>
> **On self-evaluation bias:** (1) Our pipeline uses different models for different subtasks (Gemini-3-Pro, Gemini-3-Flash, Gemini-2.5-Flash), so no single model evaluates its own outputs end-to-end; (2) GPT-5.2 performs comparably to Gemini-3-Pro in verifiable claim identification (83.9 vs 84.2 BAcc); (3) the pipeline is validated against human annotations on both Gemini and Qwen outputs.
>
> We ran alternative evaluator configurations where subtasks 1-2 are replaced with a single model, while subtask 3 (entailment) remains Gemini-2.5-Flash (G2.5-Flash) or Gemini-3-Pro (G3-Pro) — because subtask 3 requires multimodal input: GPT cannot process video/audio frames, and Qwen models achieve significantly lower correlation (best: 57.76/58.72 F1 for Qwen-Omni Instruct/Thinking vs. 73.1 for G3-Pro). All variants achieve strong correlations (p\<0.001):
> | Evaluator (Subtask 1-2 / Subtask 3\) | Pearson | Spearman | Kendall |
> | :--- | :--- | :--- | :--- |
> | Ours (G3-Pro, G3-Flash / G2.5-Flash) | 0.860 | 0.844 | 0.685 |
> | G3-Pro | 0.879 | 0.853 | 0.697 |
> | G2.5-Flash | 0.830 | 0.836 | 0.679 |
> | GPT-5 / G2.5-Flash | 0.757 | 0.776 | 0.609 |
> We also tested whether Gemini evaluators preferentially inflate Gemini outputs (full per-model breakdown in the updated paper due to space). Our multi-model pipeline achieves **perfect rank-order agreement** (tau=1.00) with human rankings, while GPT-5 inverts the top two (tau=0.33). The G3-Pro evaluator scores its own model's outputs (40.0) *lower* than G2.5-Flash (44.7), and our metric's spread (23.4) closely tracks human spread (24.4), confirming no systematic bias.
>
> **\> Q1: Prompting method for Gemini in Figure 3?**
> Yes, Figure 3 uses the \+CITATION approach from Table 5\. The thinking effort levels (Minimal, Low, Medium, High) correspond to different “thinking budget” settings in the Gemini API; “High” corresponds to the default \+CITATION results. We will clarify in the caption.
>
> **\> Q2: Benchmark vs. evaluation framework/metric**
> MURGAT is primarily an evaluation framework built on existing datasets, but also includes a new task formulation (Section 3.1), curated human annotations (580 sentence labels, 635 decompositions, 917 entailment labels), and evaluation splits (100 examples per dataset). We will revise language to “evaluation framework” where appropriate.
>
> **\> Q3: Refine introduction**
> Thank you for your suggestion. We will restructure the introduction to: (1) lead with our finding that MLLMs hallucinate citations even when answering correctly, (2) clearly state the contribution is an evaluation framework and metric, and (3) position the use of existing datasets as a deliberate choice to isolate attribution evaluation from data collection.

---

> > ### Author Rebuttal · Reviewer_3sXw · 2026-04-01
> >
> > I thank the authors for their detailed rebuttal. However, the rebuttal doesn't address my core concerns.
> >
> > A simple natural sentence may contain multiple sub-facts and multiple sources. If a particular sub-fact can be derived solely from departmental sources (while other sources may correspond to some other sub-facts, but are unnecessary for this specific one), it shouldn't be penalized for that. Because in our daily natural expressions, it's impossible for every sentence to consist of only one single fact and one single source. So I think there ire flaws in the evaluation metric. However, the authors didn't directly answer the question, but instead said that's how it was designed.

---

> > > ### Author Response · Authors · 2026-04-02
> > >
> > > We thank the reviewer for the clarification. We believe the concern is that when a sentence contains multiple sub-facts supported by different sources, our citation propagation unfairly penalizes sub-facts for inheriting citations meant for sibling facts. We want to clarify that (1) this only applies to multi-citation sentences (14.8% of our data), (2) our pipeline already includes a Splitting rule and human annotation protocol designed for exactly this case, and (3) new empirical analysis shows the metric behaves appropriately in both inline and end-dumped citation patterns.
> > >
> > > **Our pipeline already accounts for multi-source sentences.** The decomposition prompt includes a **Splitting rule** (Figure 4, L935–990): *"If citations are embedded (e.g., 'A (visual, 1:00) hits B (visual, 2:00)'), assign the specific citation only to the relevant fact."* When citations are placed inline, the decomposer assigns them to the correct sub-facts. Our human annotators similarly perform fine-grained citation-to-fact assignment (Appendix A.2). A real example from our data:
> > >
> > > > "The nucleolus 'creates rRNA to make Ribosomes' **(audio, 1:02–1:06)** and 'sends mRNA to Ribosome to make proteins' **(visual, 1:30–1:39; audio, 1:30–1:39)**."
> > >
> > > The rRNA facts inherit only (audio, 1:02–1:06); mRNA facts inherit (visual+audio, 1:30–1:39). **Precision 1.0, Recall 1.0** — no penalty despite multiple facts and sources.
> > >
> > > In Figure 1, citations are instead grouped at the end, and so the Splitting rule propagate to all facts. Subtask 3 (Section 3.2, L179–198) then verifies each and finds the visual citation does not support F2, resulting in a precision penalty. Thus, the penalty reflects the model's citation placement, not a metric flaw.
> > >
> > > **Empirical analysis.** We automatically classified all 600 annotated sentences by detecting citation patterns via regex matching and checking whether multiple citations appear interleaved within the sentence text (inline) or clustered at the boundary (end-dumped). Summary: only 14.8% of sentences have multiple citations; of those, 79.8% are end-dumped and 20.2% inline.
> > >
> > > | Model | Total | Multi-cite | Inline | End-dumped | % End-dumped |
> > > | :---- | :---- | :---- | :---- | :---- | :---- |
> > > | Gemini-2.5-Flash | 225 | 58 (25.8%) | 11 | 47 | 81.0% |
> > > | Gemini-3-Pro | 127 | 21 (16.5%) | 6 | 15 | 71.4% |
> > > | Qwen3-Omni-Inst. | 136 | 5 (3.7%) | 0 | 5 | 100.0% |
> > > | Qwen3-Omni-Think. | 112 | 5 (4.5%) | 1 | 4 | 80.0% |
> > > | **All** | **600** | **89 (14.8%)** | **18** | **71** | **79.8%** |
> > >
> > > End-dumping is the dominant pattern (79.8%). However, examining our existing human annotations (Appendix A), across all 71 end-dumped sentences, annotators assigned the full citation set to every atomic fact — judging each citation as applicable to all facts, not just a subset. This indicates that when models group citations at the end, the cited segments tend to broadly cover the full sentence, making blanket propagation appropriate.
> > >
> > > The 18 inline cases show that when fact-specific mapping is needed, models place citations near the relevant claims and the Splitting rule handles them correctly. This two-regime pattern suggests the metric captures a meaningful distinction in citation quality. The effect is also **symmetric across models** (71–100% end-dumping), so rankings are unaffected, and **recall is entirely unaffected**.
> > >
> > > We will add this analysis and discussion to the revised paper, and are happy to address any remaining questions.

---

### Decision · Program_Chairs · 2026-04-30

**Decision:**

Accept (regular)

**Comment:**

The paper introduces MURGAT, an evaluation framework designed to assess fact-level multimodal attribution in large language models across video, audio, and graphical inputs.
The core contribution lies in its three-stage automated pipeline that decomposes model responses into atomic facts and verifies their corresponding temporal citations.

The reviewers generally agree on the importance and novelty of the task, noting that existing benchmarks fail to capture the complexities of multi-step multimodal reasoning and temporal grounding. The authors successfully addressed several critical concerns during the rebuttal phase. Specifically, they expanded their evaluation to include a wider variety of models, demonstrating that attribution failures are a systemic issue across different model families rather than being limited to Gemini. They also provided robust statistical evidence through bootstrap significance testing to defend the reliability of their automated metric and clarified how their decomposition rules handle complex, multi-source sentences to avoid unfair penalties.

While some reviewers remained cautious regarding the sample size used for human validation and the inherent challenges of using LLMs as judges without ground truth, the authors mitigated these concerns by demonstrating high correlation with human judgments and providing a clear verification protocol against source evidence. The paper is technically sound and provides actionable insights into the disconnect between internal reasoning and verifiable attribution in MLLMs.

Given the strength of the empirical findings and the utility of the proposed evaluation framework for the community, I recommend a Weak Accept, but I would not mind if my recommendation was bumped down.